# Generalization and Optimization of SGD with Lookahead

## Abstract

The Lookahead optimizer (Zhang et al., 2019) enhances deep learning models by employing a dual-weight update mechanism, which has been shown to improve the performance of underlying optimizers such as SGD. However, most theoretical studies focus on its convergence on training data, leaving its generalization capabilities less understood. Existing generalization analyses are often limited by restrictive assumptions, such as requiring the loss function to be globally Lipschitz continuous, and their bounds do not fully capture the relationship between optimization and generalization. In this paper, we address these issues by conducting a rigorous stability and generalization analysis of the Lookahead optimizer with minibatch SGD. We leverage on-average model stability to derive generalization bounds for both convex and strongly convex problems without the restrictive Lipschitzness assumption. Our analysis demonstrates a linear speedup with respect to the batch size in the convex setting.

## 1 Introduction

Stochastic optimization has become the method of choice to train modern machine learning models due to its efficiency and scalability (Kingma & Ba, 2014). A simple stochastic optimization method is the minibatch stochastic gradient descent (minibatch SGD) (Cotter et al., 2011b; Dekel et al., 2012; Li et al., 2014; Shamir & Srebro, 2014), where a minibatch of training examples are randomly sampled to build gradient estimates with a reduced variance. Due to its simplicity, computational efficiency and strong generalization in practice (Zhou et al., 2020; Bottou et al., 2018), minibatch SGD remains one of the most preferable algorithms. Another representative stochastic optimization method is Adam (Kingma & Ba, 2014), which augments SGD with coordinate-wise adaptive learning rates and momentum, often accelerating convergence and improving robustness to ill-conditioning.

To further enhance generalization performance, the Lookahead optimizer (Zhang et al., 2019) was introduced as an orthogonal method. It introduces a two-timescale updating framework of two parameters: the fast weights $\mathbf{v}$ and the slow weights $\mathbf{w}$. In the inner loop, starting from the slow weights $\mathbf{w}$, the fast weights are updated by applying a standard optimizer $\mathcal{A}$ for $k$ times and output $\mathbf{v}_k$; for the outer loop, the slow weights are updated towards the fast weights by $\mathbf{w}_+ = \alpha \mathbf{v}_k + (1 - \alpha)\mathbf{w}$, where $\alpha \in (0, 1]$ is an interpolation parameter. This mechanism dampens oscillations, reduces sensitivity to learning-rate schedules and synchronization periods, and improves robustness across tasks with negligible overhead, often matching or improving the accuracy of the underlying base optimizer (Zhang et al., 2019).

The empirical efficiency of the Lookahead optimizer motivates a lot of theoretical studies to understand its behavior. However, most of existing studies focus on their convergence to minimize the training errors (Yang et al., 2024; Chen et al., 2022b; Zhang et al., 2019). As a comparison, there are far less studies on how the training behavior generalizes to testing examples, which is a concept of central interest in machine learning. To our best knowledge, the only work on the generalization analysis is Zhou et al. (2021), which conducted a stability analysis to argue that the Lookahead optimizer can generalize better than SGD and Adam. While these results provide a sound foundation on the use of the Lookahead mechanism, there are still some issues to be addressed. For example, their analysis hinges on the Lipschitzness condition on the loss, which is often restrictive in high-dimensional problems where gradients can be unbounded and the loss landscapes are non-Lipschitz globally.

Furthermore, their stability bounds are not optimistic and cannot fully capture the connection between generalization and optimization.

This paper aims to address the above issues by improving the existing stability and generalization analysis of the Lookahead optimizer. Our main contributions can be summarized as follows.

1. We leverage the on-average model stability to analyze the generalization behavior of the Lookahead methods for both convex and strongly convex problems. Our analysis removes the restrictive Lipschitzness assumptions of the loss functions, which can imply effective generalization bounds in the case with unbounded gradients. Furthermore, our analysis clearly shows how the interpolation parameter $\alpha$ strengthens the stability, which shows a clear benefit of the Lookahead mechanism.

2. Our stability bounds are optimistic, meaning that they depend on the empirical risk of the iterates produced by the algorithm. As the optimizer minimizes the empirical risk during the optimization process, our bounds become progressively tighter, offering a more refined and practical characterization of stability compared to existing bounds that rely on worst-case global constants.

3. By carefully combining our stability bounds with the convergence rates, we establish optimal excess risk rates for SGD with Lookahead. We show that it achieves a rate of $\mathcal{O}(1/n)$ for convex problems and a rate of $\mathcal{O}(1/(n\mu))$ for $\mu$-strongly convex problems, where $n$ is the sample size. Furthermore, our analysis shows a linear speedup with respect to the batch size $b$, meaning that the number of required iterations is decreased by a factor of $b$ to achieve the optimal excess risk bounds.

The paper is organized as follows. We review the related work in Section 2 and introduce the problem formulation in Section 3. We present our main theoretical results in Section 5. The detailed proofs are provided in Appendix A. We conclude the paper in Section 6.

## 2 RELATED WORK

**Stability and Generalization Analysis** A central challenge in machine learning is ensuring that models generalize well from finite training data to unseen examples. Algorithmic stability is an effective concept to study the generalization gap of learning algorithms, which can incorporate the special property of learning algorithms to derive algorithm-dependent generalization bounds (Bousquet & Elisseeff, 2002). A most widely used stability measure is the uniform stability, which is frequently used to analyze the generalization of regularization methods (Bousquet & Elisseeff, 2002) and stochastic optimization methods (Hardt et al., 2016). This stability concept was relaxed to on-average stability and on-average model stability to derive data-dependent generalization bounds (Shalev-Shwartz et al., 2010; Kuzborskij & Lampert, 2018; Lei & Ying, 2020; Schliserman & Koren, 2022). Recently, algorithm stability has found very successful applications in understanding the generalization behavior of complex models and training paradigms, including zeroth-order SGD (Nikolakakis et al., 2022; Chen et al., 2023), differential privacy (Bassily et al., 2019; 2020), asynchronous SGD (Deng et al., 2025) and neural network training (Richards & Kuzborskij, 2021; Wang et al., 2025a; Taheri & Thrampoulidis, 2024; Deora et al., 2024).

**Lookahead Optimizer** The Lookahead optimizer (Zhang et al., 2019) represents a significant advancement in optimization techniques for deep learning by employing a dual-weight update mechanism that separates "fast weights" (updated via a base optimizer) and "slow weights" (updated through exponential moving averaging). It reduces sensitivity to hyperparameters such as learning rates and synchronization periods, making it particularly robust in complex training scenarios where conventional optimizers struggle with oscillation or divergence (Nag, 2020; Zuo et al., 2024). Lookahead is widely adopted and extended across diverse domains including online learning (Chen et al., 2022a), aircraft maintenance scheduling (Deng & Santos, 2022), reinforcement learning (Merlis, 2024; Winnicki et al., 2025; Zhang et al., 2025), precision path tracking (Wang et al., 2025b), and healthcare prediction (Chen et al., 2022c; Adeshina & Adedigba, 2022). Various algorithmic extensions for Lookahead have also been introduced, including Multilayer Lookahead (Pushkin & Barba, 2021), Sharpness-Aware Lookahead (SALA) (Tan et al., 2024), Multi-step Lookahead Bayesian Optimization (Byun et al., 2022), and Lookaround Optimizer (Zhang et al., 2023).

## 3 NOTATIONS AND PRELIMINARIES

Let $\mathcal{D}$ be a probability measure defined on a sample space $\mathcal{Z} = \mathcal{X} \times \mathcal{Y}$, where $\mathcal{X}$ is an input space and $\mathcal{Y}$ is an output space. Let $S = \{z_1, z_2, \ldots, z_n\}$ be a sample drawn independently and identically (i.i.d.) from $\mathcal{D}$, based on which we aim to learn a model $h : \mathcal{X} \mapsto \mathbb{R}$ for prediction. We assume the model is characterized by a parameter $\mathbf{w} \in \mathcal{W} \subseteq \mathbb{R}^d$, where $\mathcal{W}$ is a parameter space. The performance of a model $\mathbf{w}$ on a single data point $z$ is measured by a non-negative loss function $f(\mathbf{w}; z)$, from which we can define empirical risks $F_S(\mathbf{w})$ and population risks $F(\mathbf{w})$ to measure the behavior of $\mathbf{w}$ on training and testing datasets, respectively

$$F_S(\mathbf{w}) := \frac{1}{n} \sum_{i=1}^{n} f(\mathbf{w}; z_i) \quad \text{and} \quad F(\mathbf{w}) := \mathbb{E}_{z \sim \mathcal{D}}[f(\mathbf{w}; z)],$$

where $\mathbb{E}_z[\cdot]$ means the expectation w.r.t. $z$.

We often apply a randomized optimizer $A$ to approximately minimize $F_S$ to train a model. We use $A(S)$ to denote the model produced by applying $A$ to $S$, and are interested in its relative performance w.r.t. the best model $\mathbf{w}^* = \arg\min_{\mathbf{w} \in \mathcal{W}} F(\mathbf{w})$, which is quantified by the excess risk defined by $\mathbb{E}[F(A(S)) - F(\mathbf{w}^*)]$. A powerful method to study the excess risk is to decompose it into two components (Bousquet & Bottou, 2008):

$$\mathbb{E}[F(A(S)) - F(\mathbf{w}^*)] = \underbrace{\mathbb{E}[F(A(S)) - F_S(A(S))]}_{\text{Generalization Error}} + \underbrace{\mathbb{E}[F_S(A(S)) - F_S(\mathbf{w}^*)]}_{\text{Optimization Error}}, \quad (3.1)$$

where the expectation is taken over the randomness of the training set $S$ and any randomness within the algorithm itself. Here we use the identity $\mathbb{E}[F_S(\mathbf{w}^*)] = F(\mathbf{w}^*)$. We refer to $\mathbb{E}[F(A(S)) - F_S(A(S))]$ as the generalization gap, which shows the cost we suffer when we generalize the behavior from training to testing. A small generalization gap indicates that the model does not overfit the training data and its performance is likely to be representative of its true performance. We refer to $\mathbb{E}[F_S(A(S)) - F_S(\mathbf{w}^*)]$ as the optimization error, which measures the gap between the estimated model and the true optimal model on empirical risk.

We introduce the following necessary definitions for our analysis. Let $\|\cdot\|_2$ denote the Euclidean norm.

**Definition 1.** Let $g : \mathcal{W} \mapsto \mathbb{R}$, $G, L > 0$ and $\mu \geq 0$. We denote the gradient of $g$ by $\nabla g$.

1. A function $g(\mathbf{w})$ is $\mu$-strongly convex for some $\mu > 0$ if it satisfies:

$$g(\mathbf{w}_1) \geq g(\mathbf{w}_2) + \langle \nabla g(\mathbf{w}_2), \mathbf{w}_1 - \mathbf{w}_2 \rangle + \frac{\mu}{2}\|\mathbf{w}_1 - \mathbf{w}_2\|_2^2, \quad \forall \mathbf{w}_1, \mathbf{w}_2 \in \mathcal{W}.$$

A function $g(\mathbf{w})$ is convex if it is $\mu$-strongly convex with $\mu = 0$.

2. A function $g(\mathbf{w})$ is $G$-Lipschitz continuous if the function value is bounded in its change:

$$|g(\mathbf{w}_1) - g(\mathbf{w}_2)| \leq G\|\mathbf{w}_1 - \mathbf{w}_2\|_2, \quad \forall \mathbf{w}_1, \mathbf{w}_2 \in \mathcal{W}.$$

3. A differentiable function $g(\mathbf{w})$ is $L$-smooth if its gradient is Lipschitz continuous with the constant $L$:

$$\|\nabla g(\mathbf{w}_1) - \nabla g(\mathbf{w}_2)\|_2 \leq L\|\mathbf{w}_1 - \mathbf{w}_2\|_2, \quad \forall \mathbf{w}_1, \mathbf{w}_2 \in \mathcal{W}.$$

## 4 ALGORITHMIC STABILITY

To control the generalization gap, we analyze the stability of our learning algorithm. We say an algorithm is on-average stable if its output model does not change significantly when a single data point in the training set is modified. Let $A$ be a learning algorithm that takes a dataset $S$ and outputs a model $A(S)$. We denote $S \sim S'$ if $S$ and $S'$ differ by at most one data point. Specifically, we let $S^{(i)}$ be a dataset identical to $S$ except that the $i$-th data $z_i$ is replaced with a new point $z_i'$, drawn from the same distribution $\mathcal{D}$. That is, $S^{(i)} = \{z_1, \ldots, z_{i-1}, z_i', z_{i+1}, \ldots, z_n\}$.

**Definition 2** (Uniform Stability). An algorithm $A$ has uniform stability $\epsilon$ if

$$\sup_{z \in \mathcal{Z}} \sup_{S \sim S'} \mathbb{E}\left[|f(A(S); z) - f(A(S'); z)|\right] \le \epsilon.$$

**Definition 3** (On-Average Model Stability (Lei & Ying, 2020)). We say a randomized optimizer $A$ is $\ell_1$ on-average model $\epsilon$-stable if

$$\mathbb{E}_{S,S',A}\left[\frac{1}{n}\sum_{i=1}^{n}\|A(S) - A(S^{(i)})\|_2\right] \le \epsilon.$$

We say $A$ is $\ell_2$ on-average model $\epsilon$-stable if

$$\mathbb{E}_{S,S',A}\left[\frac{1}{n}\sum_{i=1}^{n}\|A(S) - A(S^{(i)})\|_2^2\right] \le \epsilon^2.$$

The following lemma provides a connection between the generalization gap and on-average model stability.

**Lemma 1** ((Lei & Ying, 2020)). *Let $S, S'$ and $S^{(i)}$ be constructed as in Definition 2, and let $\gamma > 0$.*

*(a) Suppose for any $z$, the function $\mathbf{w} \mapsto f(\mathbf{w}; z)$ is convex. If $A$ is $\ell_1$ on-average model $\epsilon$-stable and $\sup_z \|\nabla f(A(S); z)\|_2 \le G$ for any $S$, then $|\mathbb{E}_{S,A}[F_S(A(S)) - F(A(S))]| \le G\epsilon$.*

*(b) Suppose for any $z$, the function $\mathbf{w} \mapsto f(\mathbf{w}; z)$ is nonnegative and $L$-smooth. If $A$ is $\ell_2$ on-average model $\epsilon$-stable, then the following inequality holds*

$$\mathbb{E}_{S,A}[F(A(S)) - F_S(A(S))] \le \frac{L}{\gamma}\mathbb{E}_{S,A}[F_S(A(S))] + \frac{L+\gamma}{2n}\sum_{i=1}^{n}\mathbb{E}_{S,S',A}[\|A(S^{(i)}) - A(S)\|_2^2].$$

### 4.1 LOOKAHEAD OPTIMIZER

The Lookahead optimizer (Zhang et al., 2019), detailed in Algorithm 1, employs a two-loop structure: an inner loop to update fast weights, and an outer loop to update slow weights. In the inner loop, a standard optimizer $\mathcal{A}$ (e.g. SGD or Adam) starts from the previous slow weight model $\mathbf{w}_{t-1}$ and updates fast weights $\mathbf{v}_{k,t}$ with appropriate inner step sizes $\eta_{\tau,t}$ for $k$ iterations. In the $t$-th iteration of the outer loop, the fast weight model $\mathbf{v}_{k,t}$ is then used to update the slow weight model via a linear interpolation

$$\mathbf{w}_t = (1 - \alpha)\mathbf{w}_{t-1} + \alpha\mathbf{v}_{k,t} \tag{4.1}$$

where $\alpha \in (0, 1)$ is the outer step size.

---

**Algorithm 1** Lookahead Optimizer

1: **Inputs:** Data set $\mathcal{S}$, initial model $\mathbf{w}_0$, base optimizer $\mathcal{A}$, fast-weight step number $k$ and learning rates $\{\{\eta_{\tau,t}\}_{\tau=0}^{k-1}\}_{t=1}^{T}$, slow-weight step number $T$ and learning rate $\alpha \in (0, 1)$.
2: **for** $t = 1, 2, \ldots, T$ **do**
3:     $\mathbf{v}_{0,t} = \mathbf{w}_{t-1}$
4:     **for** $\tau = 1, 2, \ldots, k$ **do**
5:         $\mathbf{v}_{\tau,t} = \mathcal{A}(\mathbf{v}_{\tau-1,t}, \eta_{\tau-1,t}, \mathcal{S})$
6:     **end for**
7:     $\mathbf{w}_t = (1 - \alpha)\mathbf{w}_{t-1} + \alpha\mathbf{v}_{k,t}$
8: **end for**
9: **Outputs:** Slow model $\mathbf{w}_T$

---

We use minibatch SGD as the standard optimizer $\mathcal{A}$, which is widely used in deep learning. The inner loop is then reformulated as in Algorithm 2. At the $\tau$'th iteration, SGD collects a minibatch $\mathcal{B}_{\tau,t}$ by randomly drawing $|\mathcal{B}_{\tau,t}|$ data points from $\mathcal{S}$ independently, where $|\cdot|$ denotes the cardinality. Then it updates $\{\mathbf{v}_{\tau,t}\}_{\tau=1}^{k}$ by

$$\mathbf{v}_{\tau,t} = \mathbf{v}_{\tau-1,t} - \frac{\eta_{\tau-1,t}}{|\mathcal{B}_{\tau,t}|}\sum_{z \in \mathcal{B}_{\tau,t}}\nabla f(\mathbf{v}_{\tau-1,t}; z),$$

where $\eta_{\tau,t}$ is a positive step size.

---

**Algorithm 2** Stochastic Gradient Descent (SGD)

---

1: **Inputs:** Data set $\mathcal{S}$, learning rates $\{\eta_{\tau,t}\}_{\tau=0}^{k-1}$, initial model $\mathbf{v}_{0,t}$,
2: **for** $\tau = 1, 2, \ldots, k$ **do**
3: $\quad \mathbf{v}_{\tau,t} = \mathbf{v}_{\tau-1,t} - \frac{\eta_{\tau-1,t}}{|\mathcal{B}_{\tau,t}|} \sum_{z \in \mathcal{B}_{\tau,t}} \nabla f(\mathbf{v}_{\tau-1,t}; z)$
4: **end for**
5: **Outputs:** Fast model $\mathbf{v}_{k,t}$

---

## 5 GENERALIZATION ANALYSIS OF LOOKAHEAD ALGORITHM

In this section, we discuss the stability performance of Lookahead on convex and strongly convex problems. While previous work has shown that Lookahead achieves lower excess risk error compared to its vanilla inner optimizer when choosing $\mathcal{A}$ as SGD (Zhou et al., 2021), existing analysis of its generalization and optimization error suffer from two key limitations. First, they hinge on a restrictive Lipschitzness condition on the loss function. Second, they cannot imply optimistic rates to show the benefit of low-noise condition to get fast rates. In the following sections, we will analyze the stability bound of Lookahead via the $\ell_2$ on-average model stability. This approach notably allows us to derive generalization bounds for Lookahead without requiring the Lipschitzness condition (Lei et al., 2025). Furthermore, by carefully selecting the algorithm's hyperparameters, we establish optimal excess risk bounds.

### 5.1 CONVEX CASE

We first investigate stability bounds of Lookahead under convex condition, where Eq. (5.1) considers the $\ell_1$ on-average stability and Eq. (5.2) considers the $\ell_2$ on-average stability. The proof will be given in Appendix A.1.

**Theorem 2** (Stability Bound of Lookahead: Convex Case). *Suppose the map $\mathbf{w} \mapsto f(\mathbf{w}; z)$ is convex, nonnegative and $L$-smooth for all $z \in \mathcal{Z}$. Let $\{\mathbf{v}_{\tau,t}\}$ and $\{\mathbf{w}_t\}$, $\{\mathbf{v}_{\tau,t}^{(i)}\}$ and $\{\mathbf{w}_t^{(i)}\}$ be produced based on $S$ and $S^{(i)}$ respectively with $\eta_{\tau,t} \leq \frac{1}{L}$. We have*

$$\frac{1}{n} \sum_{i=1}^{n} \mathbb{E}\big[\|\mathbf{w}_{t+1} - \mathbf{w}_{t+1}^{(i)}\|_2\big] \leq \alpha \sum_{h=1}^{t+1} \sum_{j=1}^{k-1} \frac{2\eta_{j,h}\sqrt{2L\mathbb{E}\left[F_S\left(\mathbf{v}_{j,h}\right)\right]}}{n} \tag{5.1}$$

*and*

$$\frac{1}{n} \sum_{i=1}^{n} \mathbb{E}\big[\|\mathbf{w}_{t+1} - \mathbf{w}_{t+1}^{(i)}\|_2^2\big] \leq \left(\frac{16\alpha^2 L}{nb} + \frac{16\alpha^2 L(t+1)k}{n^2}\right) \sum_{h=1}^{t+1} \sum_{j=1}^{k-1} \eta_{j,h}^2 \mathbb{E}\left[F_S\left(\mathbf{v}_{j,h}\right)\right]. \tag{5.2}$$

**Remark 1** (Comparison with existing stability bounds for Lookahead). For $L$-smooth, $G$-Lipschitz and convex problems, a similar $\ell_1$-stability bound was derived in (Zhou et al., 2021) as shown below

$$\frac{1}{n} \sum_{i=1}^{n} \mathbb{E}\big[\|\mathbf{w}_{t+1} - \mathbf{w}_{t+1}^{(i)}\|_2\big] \leq \frac{2\alpha\eta GkT}{n}.$$

This bound grows linearly with $kT$, is independent of the mini-batch size $b$, and involves the global Lipschitz constant $G$. Our analysis removes the global $G$-Lipschitz requirement and thus avoids the $G$ factor. A notable feature of our bound is its dependence on the empirical risk, $\mathbb{E}[F_S(\mathbf{v}_{j,h})]$, rather than the global Lipschitz constant $G$ in (Zhou et al., 2021). Since the objective of the inner-loop optimizer is precisely to minimize $F_S$, we expect this term to decrease as training progresses. Consequently, our stability bounds become progressively tighter throughout the optimization process (Kuzborskij & Lampert, 2018; Lei & Ying, 2020). Furthermore, the bound in Eq. (5.2) provides clear intuition about the role of Lookahead's hyperparameters:

- **Batch Size** ($b$)**:** The term $1/nb$ shows that increasing the minibatch size improves stability. As a comparison, the stability analysis in (Zhou et al., 2021) does not show the effect of the batch size since their stability bound is independent of $b$.

- **Inner Loop Iteration Number** ($k$): The bound increases with $k$, suggesting that running the inner loop for too many steps can degrade stability, likely due to the fast weights overfitting to the training set $S$.

- **Outer Loop Step Size** ($\alpha$): Stability is proportional to $\alpha$. A smaller $\alpha$ dampens the influence of the potentially unstable fast weights, leading to a more stable trajectory for the slow weights. This shows a clear advantage of the Lookahead mechanism in improving the stability and generalization.

We get the generalization bound via plugging the stability bounds in Theorem 2 into Lemma 1. Together with the optimization bound in Lemma 9, we have the following excess risk bound. The proof is given in Appendix A.2. We denote $A \lesssim B$ if there exists a universal constant $C > 0$ such that $A \leq CB$. We denote $A \gtrsim B$ if there exists a universal constant $C$ such that $A \geq CB$. We denote $A \asymp B$ if $A \lesssim B$ and $A \gtrsim B$.

**Theorem 3** (Excess Risk Bound of Lookahead: Convex Case). *Let the assumptions of Theorem 2 hold and $R = Tk$. Then for $\bar{\mathbf{v}}_R = \frac{1}{Tk} \sum_{t=1}^{T} \sum_{\tau=0}^{k-1} \mathbf{v}_{\tau,t}$ and $\gamma > 0$, we have*

$$
\mathbb{E}\left[F(\bar{\mathbf{v}}_R)\right] - F(\mathbf{w}^*) \lesssim \frac{L\eta F(\mathbf{w}^*)}{b} + \frac{1}{\alpha\eta R} + \frac{F(\mathbf{w}^*) + L\eta/b + 1/(\alpha\eta R)}{\gamma}
$$
$$
+ L(L+\gamma)\alpha^2\eta^2 \left(\frac{1}{nb} + \frac{R}{n^2}\right) \left(RF(\mathbf{w}^*) + \frac{RL\eta}{b} + \frac{1}{\alpha\eta}\right). \quad (5.3)
$$

Since there are terms directly proportional to $F(\mathbf{w}^*)$, the excess risk bound will be tighter when the optimal risk $F(\mathbf{w}^*)$ is small, which is common in many machine learning problems where a model can fit the data well. Excess risk bounds with this feature are called optimistic bounds (Srebro et al., 2010). The terms involving $F(\mathbf{w}^*)$ are directly related to gradient noise, as the variance of stochastic gradients can often be bounded by the function's value at the optimum.

**Remark 2** (Comparison with Minibatch SGD). The excess risk bound for Lookahead in Theorem 3 shares a fundamental structure with the bound for Minibatch SGD as in (Lei et al., 2025). Both are optimistic bounds that explicitly depend on the optimal risk. This similarity is expected, as both analyses aim to control generalization gap by plugging stability bounds into Lemma 3.1, then adding optimization error terms. Although the structure is similar, the specific coefficients and dependencies on parameters such as $\alpha$ and the structure of the variance term differ due to the unique dynamics of the Lookahead optimizer compared to standard SGD.

We now develop an explicit excess risk bound for Lookahead by choosing step sizes and number of iterations. The proof is given in Appendix A.2.

**Corollary 4.** *Let the assumptions of Theorem 3 hold.*

1. *If $F(\mathbf{w}^*) \geq 1/n$, we can take $\eta = \frac{b}{\sqrt{nF(\mathbf{w}^*)}}$, $R \asymp \frac{n}{b}$, $\gamma = \sqrt{nF(\mathbf{w}^*)} \geq 1$, and $b \leq \sqrt{nF(\mathbf{w}^*)}/(2L)$ to derive $\mathbb{E}[F(\bar{\mathbf{v}}_R)] - F(\mathbf{w}^*) \lesssim \frac{LF(\mathbf{w}^*)^{1/2}}{\sqrt{n}} + \frac{L^2}{n}$.*

2. *If $F(\mathbf{w}^*) < 1/n$, we can take $\eta = \frac{1}{2L}$, $R \asymp n$, and $\gamma = 1$ to derive $\mathbb{E}[F(\bar{\mathbf{v}}_R)] - F(\mathbf{w}^*) \lesssim \frac{L}{n} + F(\mathbf{w}^*)$.*

**Remark 3.** Corollary 4 distinguishes between two key regimes based on the magnitude of the optimal risk $F(\mathbf{w}^*)$ relative to the sample size $n$.

1. $F(\mathbf{w}^*) \geq 1/n$: Our analysis shows that the algorithm achieves an excess risk bound of $O(\frac{1}{\sqrt{n}})$. Crucially, the number of required iterations $R$ is on the order of $n/b$, demonstrating a linear speedup (Cotter et al., 2011a). This means that by increasing the minibatch size $b$, one can use a proportionally larger learning rate $\eta$ and achieve the same error bound with fewer iterations. This acceleration is a direct benefit of variance reduction from larger batch sizes.

2. $F(\mathbf{w}^*) < 1/n$: Now the required number of iterations $R$ scales with $n$, irrespective of the batch size $b$. In this case, the linear speedup vanishes. The optimal learning rate becomes

constant, and increasing the batch size does not reduce the number of iterations needed to reach the desired error threshold. This suggests a small stochastic gradient noise, which means variance is no longer the main limitation of the learning process.

**Remark 4** (Comparison with Existing Excess Risk Bounds with Lookahead). The work (Zhou et al., 2021) gave the following excess risk bound for Lookahead under convexity and $G$-Lipschitz continuity assumption

$$\mathbb{E}[F(\bar{\mathbf{v}}_R)] - F(\mathbf{w}^*) \leq \frac{1}{2\alpha\eta kT}\mathbb{E}[\|\mathbf{w}_0 - \mathbf{w}^*\|^2] + \frac{\eta G^2}{2} + \frac{\alpha\eta G^2 kT}{n}.$$

By setting $\eta \asymp 1/\sqrt{n}$ and choosing $\alpha Tk \asymp n$, all three terms can be made to be of the order $O(1/\sqrt{n})$. This leads to an optimized excess risk bound of order $G^2/\sqrt{n}$, which is standard for stochastic convex optimization under a Lipschitz assumption. However, it is not adaptive and can be suboptimal in many practical scenarios. In the case of $F(\mathbf{w}^*) \geq 1/n$, our bound is of order $\frac{L\sqrt{F(\mathbf{w}^*)}}{\sqrt{n}}$. As the optimal risk $F(\mathbf{w}^*)$ decreases, our bound becomes tighter. For problems where $L\sqrt{F(\mathbf{w}^*)} \ll G^2$, our bound is substantially sharper than the generic $O(G^2/\sqrt{n})$ rate. In the case of $F(\mathbf{w}^*) < 1/n$, our analysis reveals a much faster convergence rate of $\lesssim \frac{L}{n}$. This is a linear convergence rate with respect to the sample size $n$. Achieving an $O(1/n)$ rate is a major acceleration compared to the standard $O(1/\sqrt{n})$ rate. It shows that Lookahead can effectively leverage low-noise conditions to converge significantly faster, a behavior that the existing bound fails to capture. Furthermore, our analysis shows a linear speedup on the batch size, while the discussions in (Zhou et al., 2021) do not show the benefit of considering minibatch in both generalization and optimization.

## 5.2 STRONGLY CONVEX CASE

We now consider strongly convex problems. The following theorem provides stability bounds for Lookahead. The proof is given in Appendix A.3.

**Theorem 5** (Stability Bound of Lookahead: Strongly Convex Case). *Suppose the map* $\mathbf{w} \mapsto f(\mathbf{w}; z)$ *is* $\mu$-*strongly convex, nonnegative and* $L$-*smooth for all* $z \in \mathcal{Z}$. *Let* $\{\mathbf{v}_{\tau,t}\}$ *and* $\{\mathbf{w}_t\}$, $\{\mathbf{v}_{\tau,t}^{(i)}\}$ *and* $\{\mathbf{w}_t^{(i)}\}$ *be produced based on* $S$ *and* $S^{(i)}$ *respectively with* $\frac{2\ln 2}{k\mu} \leq \eta_{\tau,t} \leq \frac{1}{L}$. *We have*

$$\frac{1}{n}\sum_{i=1}^{n}\mathbb{E}\big[\|\mathbf{w}_{t+1} - \mathbf{w}_{t+1}^{(i)}\|_2\big] \leq \frac{2\alpha\sqrt{2L}}{n}\sum_{t'=1}^{t+1}(1-\frac{\alpha}{2})^{t+1-t'}\sum_{j=0}^{k-1}\eta_{j,t'}\sqrt{\mathbb{E}\left[F_S\left(\mathbf{v}_{j,t'}\right)\right]}\prod_{j'=j+1}^{k-1}\left(1-\frac{\mu\eta_{j',t'}}{2}\right)$$

(5.4)

*and*

$$\frac{1}{n}\sum_{i=1}^{n}\mathbb{E}\big[\|\mathbf{w}_{t+1} - \mathbf{w}_{t+1}^{(i)}\|_2^2\big] \leq \sum_{t'=1}^{t+1}\sum_{j=0}^{k-1}\left(\frac{16\alpha^2\eta_{j,t'}^2}{nb} + \frac{32(t+1)\alpha^2\eta_{j,t'}}{n^2\mu}\right)\mathbb{E}\left[F_S\left(\mathbf{v}_{j,t'}\right)\right]\prod_{j'=j+1}^{k-1}\left(1-\frac{\mu\eta_{j',t'}}{2}\right)^2.$$

(5.5)

Eq. (5.4) provides an $\ell_1$-on-average stability bound. A key feature of this bound is its dependence on the empirical risk, $\sqrt{\mathbb{E}[F_S(\mathbf{v}_{j,t'})]}$. This indicates that the stability of the Lookahead algorithm improves as it finds iterates with smaller empirical risks. Eq. (5.5) provides an $\ell_2$-on-average stability bound. This bound explicitly shows the benefit of minibatching. The term $\frac{16\alpha^2\eta_{j,t'}}{nb}$ demonstrates that increasing the batch size $b$ directly improves the stability bound by reducing the variance introduced by the stochastic gradients. This is a crucial property for large-scale learning, confirming that larger batches contribute to a more stable training process for the Lookahead algorithm.

**Theorem 6** (Excess Risk Bound of Lookahead: Strongly Convex Case). *Let assumptions in Theorem 5 hold and let* $\eta = \frac{b\mu}{2L^2(b+1)}$, $k = \frac{2L}{\alpha\mu}$ *and* $T \asymp \log(\mu n)$, *we have*

$$\mathbb{E}[F(\mathbf{w}_T)] - F(\mathbf{w}^*) \lesssim \frac{1}{n\mu} + \big(\frac{1}{nL} + 1\big)\mathbb{E}[F_S(\mathbf{w}_S)] + \big(\frac{1}{n^2} + \frac{L}{n}\big)\mathbb{E}[\|\mathbf{w}_0 - \mathbf{w}_S\|^2]. \quad (5.6)$$

**Remark 5** (Comparison with Existing Excess Risk Bound with Lookahead). Compared with the existing Lookahead bound in the work (Zhou et al., 2021), which yields a sum of terms of order

$O(1/(\lambda^2((t+1)k)^{2\alpha})) + O(G/(n\lambda))$ and therefore requires $tk$ to scale polynomially with $n$ to reach the $O(1/n)$ regime, our Theorem 6 delivers a fast-rate excess risk of order $1/(n\mu)$ with only $T \asymp \log(\mu n)$ iterations. Moreover, our bound is adaptive: it tightens with the data through $(1/(nL)+1)\mathbb{E}[F_S(\mathbf{w}_S)]$ and through $(1/n^2 + L/n)\mathbb{E}[\|\mathbf{w}_0 - \mathbf{w}_S\|^2]$, becoming much smaller under interpolation, which is not captured by the existing result. Finally, the stepsize $\eta$ scales with the minibatch $b$, implying linear speedup in $b$, while prior analyses do not show such minibatch gains.

## 6 Conclusion

In this work, we investigate the stability and generalization properties of the Lookahead optimizer, a widely used algorithm for large-scale machine learning problems. While many discussions focus on its optimization benefits, we provide a rigorous analysis from the perspective of statistical learning theory. We develop on-average stability bounds for both convex and strongly convex problems, and we show how stability can be improved by small training errors, leading to optimistic bounds that depend on the empirical risk rather than a restrictive, global Lipschitz constant.

Our stability analysis implies optimal excess population risk bounds for both settings. Specifically, we demonstrate that Lookahead achieves the standard $O(1/\sqrt{n})$ rate for convex problems and the optimal $O(1/(n\mu))$ rate for strongly convex problems. A key finding is the adaptivity of Lookahead in the convex case, which achieves its rate without prior knowledge of the optimal risk $F(\mathbf{w}^*)$, a practical advantage over standard Minibatch SGD.

There are several limitations to our current work which open avenues for future research. A primary limitation is that our analysis is confined to convex and strongly convex loss functions. Given the prevalence of non-convex optimization in modern deep learning, extending our stability analysis to the non-convex setting is a crucial next step. Furthermore, while we establish the optimal statistical rate for the strongly convex case, our analysis does not demonstrate a linear speedup with respect to the batch size, a property observed in Minibatch SGD. Investigating whether different hyperparameter schedules could unlock such a speedup for Lookahead would be of significant interest. We plan to address these limitations in our future research.

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

## A  PROOF OF RESULTS IN SECTION 5

### A.1  PROOF OF THEOREM 2

Our proof of Theorem 2 relies on the following two lemmas. Lemma 7 shows the self-bounding property for nonnegative and smooth functions, meaning that the norm of gradients can be bounded by function values. Lemma 8 establishes the co-coercivity of smooth and convex functions, as well as the non-expansiveness of the gradient operator $\mathbf{w} \mapsto \mathbf{w} - \eta \nabla f(\mathbf{w}; z)$.

**Lemma 7** (Self-Bounding Property (Srebro et al., 2010)). *Assume for all z, the function $\mathbf{w} \mapsto f(\mathbf{w}; z)$ is nonnegative and L-smooth. Then*

$$\|\nabla f(\mathbf{w}; z)\|_2^2 \le 2L f(\mathbf{w}; z).$$

**Lemma 8** ((Hardt et al., 2016)). *Assume for all $z \in Z$, the function $\mathbf{w} \mapsto f(\mathbf{w}; z)$ is convex and L-smooth. Then for $\eta \le 2/L$ we have*

$$\|(\mathbf{w} - \eta \nabla f(\mathbf{w}; z)) - (\mathbf{w}' - \eta \nabla f(\mathbf{w}'; z))\|_2 \le \|\mathbf{w} - \mathbf{w}'\|_2.$$

*Furthermore, if $\mathbf{w} \mapsto f(\mathbf{w}; z)$ is $\mu$-strongly convex and $\eta \le 1/L$ then*

$$\|(\mathbf{w} - \eta \nabla f(\mathbf{w}; z)) - (\mathbf{w}' - \eta \nabla f(\mathbf{w}'; z))\|_2 \le (1 - \eta\mu/2) \|\mathbf{w} - \mathbf{w}'\|_2,$$
$$\|(\mathbf{w} - \eta \nabla f(\mathbf{w}; z)) - (\mathbf{w}' - \eta \nabla f(\mathbf{w}'; z))\|_2^2 \le (1 - \eta\mu) \|\mathbf{w} - \mathbf{w}'\|_2^2.$$

We can now prove Theorem 2. For simplicity, we define $J_{\tau,t} = \{i_{\tau,t}^{(1)}, \ldots, i_{\tau,t}^{(b)}\}$, where $i_{\tau,t}^{(j)} \sim \text{Unif}([n])$ is the $j$-th index sampled to compute a stochastic gradient for minibatch SGD, i.e., $\mathcal{B}_{\tau,t} = \{z_{i_{\tau,t}^{(1)}}, \ldots, z_{i_{\tau,t}^{(b)}}\}$.

*Proof.* To begin with, define

$$A_{\tau,t}^{(m)} = |\{j : i_{\tau,t}^{(j)} = m\}|,$$

that is, $A_{\tau,t}^{(m)}$ represents the number of indices equal to $m$ in the batch of $t$-th outer loop iteration, and $\tau$-th inner loop iteration. Then we can reformulate the Lookahead update as

$$
\begin{aligned}
\mathbf{w}_{t+1} &= (1-\alpha)\mathbf{w}_t + \alpha\mathbf{v}_{k,t+1} \\
&= (1-\alpha)\mathbf{w}_t + \alpha\Big(\mathbf{v}_{k-1,t+1} - \frac{\eta_{k-1,t+1}}{b}\sum_{m=1}^{n} A_{k-1,t+1}^{(m)}\nabla f(\mathbf{v}_{k-1,t+1}; z_m)\Big), \\
\mathbf{w}_{t+1}^{(i)} &= (1-\alpha)\mathbf{w}_t^{(i)} + \alpha\Big(\mathbf{v}_{k-1,t+1}^{(i)} - \frac{\eta_{k-1,t+1}}{b}\sum_{m:m\neq i}^{n} A_{k-1,t+1}^{(m)}\nabla f(\mathbf{v}_{k-1,t+1}^{(i)}; z_m) \\
&\qquad - \frac{A_{k,t+1}^{(i)}\eta_{k-1,t+1}}{b}\nabla f(\mathbf{v}_{k-1,t+1}^{(i)}; z_i')\Big),
\end{aligned}
\tag{A.1}
$$

from which we know

$$\|\mathbf{w}_{t+1} - \mathbf{w}_{t+1}^{(i)}\|_2 \leq (1-\alpha)\|\mathbf{w}_t - \mathbf{w}_t^{(i)}\|_2 + \alpha\|\mathbf{v}_{k,t+1} - \mathbf{v}_{k,t+1}^{(i)}\|_2$$

$$\leq (1-\alpha)\|\mathbf{w}_t - \mathbf{w}_t^{(i)}\|_2 + \alpha\Big\|\mathbf{v}_{k-1,t+1} - \frac{\eta_{k-1,t+1}}{b}\sum_{m:m\neq i}^{n} A_{k-1,t+1}^{(m)}\nabla f(\mathbf{v}_{k-1,t+1}; z_m)$$

$$- \frac{A_{k-1,t+1}^{(i)}\eta_{k-1,t+1}}{b}\nabla f(\mathbf{v}_{k-1,t+1}; z_i) - \mathbf{v}_{k-1,t+1}^{(i)} + \frac{\eta_{k-1,t+1}}{b}\sum_{m:m\neq i}^{n} A_{k-1,t+1}^{(m)}\nabla f(\mathbf{v}_{k-1,t+1}^{(i)}; z_m)$$

$$+ \frac{A_{k-1,t+1}^{(i)}\eta_{k-1,t+1}}{b}\nabla f(\mathbf{v}_{k-1,t+1}^{(i)}; z_i')\Big\|_2.$$

Define $\mathfrak{C}_{k-1,t+1}^{(i)} = \|\nabla f(\mathbf{v}_{k-1,t+1}; z_i) - \nabla f(\mathbf{v}_{k-1,t+1}^{(i)}; z_i')\|_2$. By assumption, $f$ is $L$-smooth and $\sum_{m:m\neq i}^{n} A_{k-1,t+1}^{(m)} \leq b$, from which we know $\mathbf{v} \mapsto \frac{1}{b}\sum_{m:m\neq i}^{n} A_{k-1,t+1}^{(m)} f(\mathbf{v}; z_m)$ is $L$-smooth. Since by assumption $\eta_{k-1,t+1} \leq \frac{1}{L}$, by Lemma 8 we have

$$\|\mathbf{w}_{t+1} - \mathbf{w}_{t+1}^{(i)}\|_2$$

$$\leq (1-\alpha)\|\mathbf{w}_t - \mathbf{w}_t^{(i)}\|_2 + \frac{\alpha A_{k-1,t+1}^{(i)}\eta_{k-1,t+1}}{b}\|\nabla f(\mathbf{v}_{k-1,t+1}; z_i) - \nabla f(\mathbf{v}_{k-1,t+1}^{(i)}; z_i')\|_2$$

$$+ \alpha\Big\|\mathbf{v}_{k-1,t+1} - \frac{\eta_{k-1,t+1}}{b}\sum_{m:m\neq i}^{n} A_{k-1,t+1}^{(m)}\nabla f(\mathbf{v}_{k-1,t+1}; z_m) - \big(\mathbf{v}_{k-1,t+1}^{(i)} - \frac{\eta_{k-1,t+1}}{b}\sum_{m:m\neq i}^{n} A_{k-1,t+1}^{(m)}\nabla f(\mathbf{v}_{k-1,t+1}^{(i)}; z_m)\big)\Big\|_2$$

$$\leq (1-\alpha)\|\mathbf{w}_t - \mathbf{w}_t^{(i)}\|_2 + \frac{\alpha\eta_{k-1,t+1}A_{k-1,t+1}^{(i)}\mathfrak{C}_{k-1,t+1}^{(i)}}{b} + \alpha\|\mathbf{v}_{k-1,t+1} - \mathbf{v}_{k-1,t+1}^{(i)}\|_2. \tag{A.2}$$

Note the above inequality actually shows a recurrent relationship on $\|\mathbf{v}_{k,t+1} - \mathbf{v}_{k,t+1}^{(i)}\|_2$ and $\|\mathbf{v}_{k-1,t+1} - \mathbf{v}_{k-1,t+1}^{(i)}\|_2$. By iteration on inner-loop, we have

$$\|\mathbf{w}_{t+1} - \mathbf{w}_{t+1}^{(i)}\|_2 \leq (1-\alpha)\|\mathbf{w}_t - \mathbf{w}_t^{(i)}\|_2 + \frac{\alpha}{b}\sum_{j=0}^{k-1}\eta_{j,t+1}A_{j,t+1}^{(i)}\mathfrak{C}_{j,t+1}^{(i)} + \alpha\|\mathbf{w}_t - \mathbf{w}_t^{(i)}\|_2$$

$$= \|\mathbf{w}_t - \mathbf{w}_t^{(i)}\|_2 + \frac{\alpha}{b}\sum_{j=0}^{k-1}\eta_{j,t+1}A_{j,t+1}^{(i)}\mathfrak{C}_{j,t+1}^{(i)},$$

where we have used that $\mathbf{v}_{0,t+1} = \mathbf{w}_t$. By iteration on outer-loop, we have

$$\|\mathbf{w}_{t+1} - \mathbf{w}_{t+1}^{(i)}\|_2 \leq \frac{\alpha}{b}\sum_{h=1}^{t+1}\sum_{j=0}^{k-1}\eta_{j,h}A_{j,h}^{(i)}\mathfrak{C}_{j,h}^{(i)}. \tag{A.3}$$

By definition of $A_{k,t}^{(m)}$, it is a random variable following the binomial distribution $B(b, \frac{1}{n})$, it then follows that

$$\mathbb{E}\big[A_{k,t}^{(m)}\big] = \frac{b}{n}, \quad \mathrm{Var}\big(A_{k,m}^{(t)}\big) = \frac{b}{n}\big(1 - \frac{1}{n}\big) \leq \frac{b}{n}. \tag{A.4}$$

Furthermore, by Lemma 7, we know

$$\mathfrak{C}_{j,h}^{(i)} \leq \|\nabla f(\mathbf{v}_{j,h}; z_i)\|_2 + \|\nabla f(\mathbf{v}_{j,h}^{(i)}; z_i')\|_2 \leq \sqrt{2Lf(\mathbf{v}_{j,h}; z_i)} + \sqrt{2Lf(\mathbf{v}_{j,h}^{(i)}; z_i')}. \tag{A.5}$$

Since $(x_i, y_i)$ and $(x_i', y_i')$ are symmetric, we know $\mathbb{E}[f(\mathbf{v}_{j,h}; z_i)] = \mathbb{E}[f(\mathbf{v}_{j,h}; z_i')]$. This, together with Eq (A.5), further implies that

$$\mathbb{E}\big[\mathfrak{C}_{j,h}^{(i)}\big] \leq 2\mathbb{E}\Big[\sqrt{2Lf(\mathbf{v}_{j,h}; z_i)}\Big]. \tag{A.6}$$

By combining (A.3) and (A.4), we have

$$\mathbb{E}\big[\|\mathbf{w}_{t+1} - \mathbf{w}_{t+1}^{(i)}\|_2\big] \leq \frac{\alpha}{b}\sum_{h=1}^{t+1}\sum_{j=0}^{k-1}\eta_{j,h}\mathbb{E}\big[A_{j,h}^{(i)}\mathfrak{C}_{j,h}^{(i)}\big] = \frac{\alpha}{b}\sum_{h=1}^{t+1}\sum_{j=0}^{k-1}\eta_{j,h}\mathbb{E}\big[\mathbb{E}_{J_{j,h}}\big[A_{j,h}^{(i)}\big]\mathfrak{C}_{j,h}^{(i)}\big]$$

$$= \frac{\alpha}{n}\sum_{h=1}^{t+1}\sum_{j=0}^{k-1}\eta_{j,h}\mathbb{E}\big[\mathfrak{C}_{j,h}^{(i)}\big] \leq \frac{2\alpha}{n}\sum_{h=1}^{t+1}\sum_{j=0}^{k-1}\eta_{j,h}\mathbb{E}\Big[\sqrt{2Lf(\mathbf{v}_{j,h}; z_i)}\Big], \tag{A.7}$$

where we used (A.6) in the last inequality. By the concavity of $x \mapsto \sqrt{x}$, we have

$$\frac{1}{n}\sum_{i=1}^{n}\mathbb{E}\big[\|\mathbf{w}_{t+1}-\mathbf{w}_{t+1}^{(i)}\|_2\big] \leq \frac{2\alpha}{n}\sum_{i=1}^{n}\sum_{h=1}^{t+1}\sum_{j=0}^{k-1}\frac{\eta_{j,h}}{n}\mathbb{E}\Big[\sqrt{2Lf(\mathbf{v}_{j,h};z_i)}\Big]$$

$$\leq \alpha\sum_{h=1}^{t+1}\sum_{j=0}^{k-1}\frac{2\eta_{j,h}}{n}\sqrt{\frac{2L}{n}\sum_{i=1}^{n}\mathbb{E}\left[f(\mathbf{v}_{j,h};z_i)\right]}$$

$$= \alpha\sum_{h=1}^{t+1}\sum_{j=0}^{k-1}\frac{2\eta_{j,h}\sqrt{2L\mathbb{E}\left[F_S(\mathbf{v}_{j,h})\right]}}{n}. \tag{A.8}$$

This established the stated $\ell_1$-stability (5.1).

To study the $\ell_2$-stability, we apply the following expectation-variance decomposition to Eq. (A.3).

$$\|\mathbf{w}_{t+1}-\mathbf{w}_{t+1}^{(i)}\|_2 \leq \frac{\alpha}{b}\sum_{h=1}^{t+1}\sum_{j=0}^{k-1}\eta_{j,h}\big(A_{j,h}^{(i)}-\frac{b}{n}\big)\mathfrak{C}_{j,h}^{(i)} + \frac{\alpha}{n}\sum_{h=1}^{t+1}\sum_{j=0}^{k-1}\eta_{j,h}\mathfrak{C}_{j,h}^{(i)}. \tag{A.9}$$

Taking square on both sides, then applying expectation with respect to $S$ and $J_{k,t}$ for $t \in [T]$ and $k \in [k]$, we have

$$\mathbb{E}\Big[\|\mathbf{w}_{t+1}-\mathbf{w}_{t+1}^{(i)}\|_2^2\Big]$$

$$\leq \frac{2\alpha^2}{b^2}\mathbb{E}\Big[\Big(\sum_{h=1}^{t+1}\sum_{j=0}^{k-1}\eta_{j,h}\big(A_{j,h}^{(i)}-\frac{b}{n}\big)\mathfrak{C}_{j,h}^{(i)}\Big)^2\Big] + \frac{2\alpha^2}{n^2}\mathbb{E}\Big[\Big(\sum_{h=1}^{t+1}\sum_{j=0}^{k-1}\eta_{j,h}\mathfrak{C}_{j,h}^{(i)}\Big)^2\Big]$$

$$= \frac{2\alpha^2}{b^2}\mathbb{E}\Big[\sum_{h,h'=1}^{t+1}\sum_{j,j'=0}^{k-1}\eta_{j,h}\eta_{j',h'}\big(A_{j,h}^{(i)}-\frac{b}{n}\big)\big(A_{j',h'}^{(i)}-\frac{b}{n}\big)\mathfrak{C}_{j,h}^{(i)}\mathfrak{C}_{j',h'}^{(i)}\Big] + \frac{2\alpha^2}{n^2}\mathbb{E}\Big[\Big(\sum_{h=1}^{t+1}\sum_{j=0}^{k-1}\eta_{j,h}\mathfrak{C}_{j,h}^{(i)}\Big)^2\Big], \tag{A.10}$$

where we have used $(a+b)^2 \leq 2(a^2+b^2)$. Note that if $(h,j) \neq (h',j')$, then (we can assume $h < h', j < j'$ without loss of generality)

$$\mathbb{E}\Big[\big(A_{j,h}^{(i)}-\frac{b}{n}\big)\big(A_{j',h'}^{(i)}-\frac{b}{n}\big)\mathfrak{C}_{j,h}^{(i)}\mathfrak{C}_{j',h'}^{(i)}\Big] = \mathbb{E}\mathbb{E}_{J_{j',h'}}\Big[\big(A_{j,h}^{(i)}-\frac{b}{n}\big)\big(A_{j',h'}^{(i)}-\frac{b}{n}\big)\mathfrak{C}_{j,h}^{(i)}\mathfrak{C}_{j',h'}^{(i)}\Big]$$

$$= \mathbb{E}\Big[\big(A_{j,h}^{(i)}-\frac{b}{n}\big)\mathbb{E}_{J_{j',h'}}\Big[A_{j',h'}^{(i)}-\frac{b}{n}\Big]\mathfrak{C}_{j,h}^{(i)}\mathfrak{C}_{j',h'}^{(i)}\Big] = 0, \tag{A.11}$$

where we notice $A_{j,h}^{(i)}$, $\mathfrak{C}_{j,h}^{(i)}$, and $\mathfrak{C}_{j',h'}^{(i)}$ are independent of $J_{j',h'}$. It then follows that

$$\mathbb{E}\big[\|\mathbf{w}_{t+1}-\mathbf{w}_{t+1}^{(i)}\|_2^2\big] \leq \frac{2\alpha^2}{b^2}\mathbb{E}\Big[\sum_{h=1}^{t+1}\sum_{j=0}^{k-1}\eta_{j,h}^2\big(A_{j,h}^{(i)}-\frac{b}{n}\big)^2\big(\mathfrak{C}_{j,h}^{(i)}\big)^2\Big] + \frac{2\alpha^2}{n^2}\mathbb{E}\Big[\Big(\sum_{h=1}^{t+1}\sum_{j=0}^{k-1}\eta_{j,h}\mathfrak{C}_{j,h}^{(i)}\Big)^2\Big]$$

$$= \frac{2\alpha^2}{b^2}\mathbb{E}\Big[\sum_{h=1}^{t+1}\sum_{j=0}^{k-1}\eta_{j,h}^2\,\mathrm{Var}\,\big(A_{j,h}^{(i)}\big)\big(\mathfrak{C}_{j,h}^{(i)}\big)^2\Big] + \frac{2\alpha^2}{n^2}\mathbb{E}\Big[\Big(\sum_{h=1}^{t+1}\sum_{j=0}^{k-1}\eta_{j,h}\mathfrak{C}_{j,h}^{(i)}\Big)^2\Big]$$

$$\leq \frac{2\alpha^2}{nb}\mathbb{E}\Big[\sum_{h=1}^{t+1}\sum_{j=0}^{k-1}\eta_{j,h}^2\big(\mathfrak{C}_{j,h}^{(i)}\big)^2\Big] + \frac{8\alpha^2}{n^2}\mathbb{E}\Big[\Big(\sum_{h=1}^{t+1}\sum_{j=0}^{k-1}\eta_{j,h}\|\nabla f(\mathbf{v}_{j,h};z_i)\|_2\Big)^2\Big],$$

where we used $\mathrm{Var}(A_{j,h}^{(i)}) = \frac{b}{n}(1-\frac{1}{n}) \leq \frac{b}{n}$ in the second inequality and used the fact that

$$\mathbb{E}\Big[\Big(\sum_{h=1}^{t+1}\sum_{j=0}^{k-1}\eta_{j,h}\mathfrak{C}_{j,h}^{(i)}\Big)^2\Big] \leq 2\mathbb{E}\Big[\Big(\sum_{h=1}^{t+1}\sum_{j=0}^{k-1}\eta_{j,h}\|\nabla f(\mathbf{v}_{j,h};z_i)\|_2\Big)^2\Big] + 2\mathbb{E}\Big[\Big(\sum_{h=1}^{t+1}\sum_{j=0}^{k-1}\eta_{j,h}\|\nabla f(\mathbf{v}_{j,h}^{(i)};z_i')\|_2\Big)^2\Big]$$

$$= 4\mathbb{E}\Big[\Big(\sum_{h=1}^{t+1}\sum_{j=0}^{k-1}\eta_{j,h}\|\nabla f\left(\mathbf{v}_{j,h};z_i\right)\|_2\Big)^2\Big].$$

We also notice that

$$\mathbb{E}\big[\big(\mathfrak{C}_{j,h}^{(i)}\big)^2\big] \leq 2\mathbb{E}\big[\|\nabla f(\mathbf{v}_{j,h},z_i)\|_2^2\big] + 2\mathbb{E}\big[\|\nabla f(\mathbf{v}_{j,h}^{(i)};z_i')\|_2^2\big]$$

$$\leq 4L\mathbb{E}\Big[f\left(\mathbf{v}_{j,h};z_i\right) + f\left(\mathbf{v}_{j,h}^{(i)};z_i'\right)\Big] = 8L\mathbb{E}\left[f\left(\mathbf{v}_{j,h};z_i\right)\right]. \tag{A.12}$$

It then follows that

$$\mathbb{E}\big[\|\mathbf{w}_{t+1} - \mathbf{w}_{t+1}^{(i)}\|_2^2\big] \leq \frac{16\alpha^2 L}{nb} \sum_{h=1}^{t+1} \sum_{j=0}^{k-1} \eta_{j,h}^2 \mathbb{E}\left[f\left(\mathbf{v}_{j,h}; z_i\right)\right] + \frac{8\alpha^2}{n^2} \mathbb{E}\Big[\Big(\sum_{h=1}^{t+1} \sum_{j=0}^{k-1} \eta_{j,h} \|\nabla f\left(\mathbf{v}_{j,h}; z_i\right)\|_2\Big)^2\Big]. \tag{A.13}$$

By taking an average over all $i \in [n]$, we have

$$\frac{1}{n} \sum_{i=1}^n \mathbb{E}\left[\|\mathbf{w}_{t+1} - \mathbf{w}_{t+1}^{(i)}\|_2^2\right]$$

$$\leq \frac{16\alpha^2 L}{n^2 b} \sum_{h=1}^{t+1} \sum_{j=0}^{k-1} \sum_{i=1}^n \eta_{j,h}^2 \mathbb{E}\left[f\left(\mathbf{v}_{j,h}; z_i\right)\right] + \frac{8\alpha^2}{n^3} \sum_{i=1}^n \mathbb{E}\Big[\Big(\sum_{h=1}^{t+1} \sum_{j=0}^{k-1} \eta_{j,h} \|\nabla f\left(\mathbf{v}_{j,h}; z_i\right)\|_2\Big)^2\Big]$$

$$\leq \frac{16\alpha^2 L}{nb} \sum_{h=1}^{t+1} \sum_{j=0}^{k-1} \eta_{j,h}^2 \mathbb{E}\left[F_S\left(\mathbf{v}_{j,h}\right)\right] + \frac{8(t+1)k\alpha^2}{n^3} \sum_{i=1}^n \sum_{h=1}^{t+1} \sum_{j=0}^{k-1} \eta_{j,h}^2 \mathbb{E}\left[\|\nabla f\left(\mathbf{v}_{j,h}; z_i\right)\|_2^2\right]$$

$$\leq \Big(\frac{16\alpha^2 L}{nb} + \frac{16\alpha^2 L(t+1)k}{n^2}\Big) \sum_{h=1}^{t+1} \sum_{j=0}^{k-1} \eta_{j,h}^2 \mathbb{E}\left[F_S\left(\mathbf{v}_{j,h}\right)\right], \tag{A.14}$$

where the second inequality holds by applying Cauchy-Schwarz inequality, and the third inequality follows from self-bounding property. The proof is completed. □

## A.2 PROOF OF THEOREM 3

We first introduce the optimization error bound for Lookahead in the convex case.

**Lemma 9** (Optimization Errors of Lookahead: Convex Case). *Suppose the assumptions in Theorem 2 hold, and further assume that $\eta < \frac{b}{L(b+1)}$, then the following inequality holds*

$$\mathbb{E}\left[F_S\left(\bar{\mathbf{v}}_R\right) - F_S\left(\mathbf{w}^*\right)\right] \leq \frac{b \mathbb{E}\left[\|\mathbf{w}_0 - \mathbf{w}_S\|^2\right]}{2\alpha\eta kT\left(b - L\eta(b+1)\right)} + \frac{L\eta \mathbb{E}\left[F_S\left(\mathbf{w}_S\right)\right]}{b - L\eta(b+1)}, \tag{A.15}$$

*where $\bar{\mathbf{v}}_R = \frac{1}{Tk} \sum_{t=1}^T \sum_{\tau=0}^{k-1} \mathbf{v}_{\tau,t}$.*

We need the following property for the $L$-smooth and convex functions for the proof.

**Lemma 10** ((Woodworth et al., 2020)). *For any $L$-smooth and convex $F$, and any $x$, and $y$,*

$$\|\nabla F\left(x\right) - \nabla F\left(y\right)\|^2 \leq L\langle\nabla F\left(x\right) - \nabla F\left(y\right), x - y\rangle,$$

*and*

$$\|\nabla F\left(x\right) - \nabla F\left(y\right)\|^2 \leq 2L\left(F\left(x\right) - F\left(y\right) - \langle\nabla F\left(y\right), x - y\rangle\right).$$

*Proof of Lemma 9.* Since $F_S(\mathbf{w}_S) \leq F_S(\mathbf{w}^*)$, an upper bound for $F_S(\bar{\mathbf{v}}_R) - F_S(\mathbf{w}_S)$ is also an upper bound for $F_S(\bar{\mathbf{v}}_R) - F_S(\mathbf{w}^*)$. For the proof below, we assume that the learning rate is constant, that is, $\eta_{\tau,t} = \eta$. We denote $B_{k,t} = \{z_{i_{k,t}^{(1)}}, \dots, z_{i_{k,t}^{(b)}}\}$ and $f\left(\mathbf{v}; B_{k,t}\right) = \frac{1}{b} \sum_{j=1}^b f(\mathbf{v}; z_{i_{k,t}^{(j)}})$. We can hence reformulate the minibatch SGD update as

$$\mathbf{v}_{\tau+1,t} = \mathbf{v}_{\tau,t} - \eta\nabla f\left(\mathbf{v}_{\tau,t}; B_{\tau,t}\right).$$

We first notice that

$$\mathbb{E}\big[\|\nabla f\left(\mathbf{v}_{\tau,t}; B_{\tau,t}\right)\|^2\big] = \mathbb{E}\big[\|\nabla f\left(\mathbf{v}_{\tau,t}; B_{\tau,t}\right) - \nabla F_S\left(\mathbf{v}_{\tau,t}\right)\|^2\big] + \mathbb{E}\big[\|\nabla F_S\left(\mathbf{v}_{\tau,t}\right)\|^2\big]$$

$$= \frac{1}{b}\mathbb{E}\big[\|\nabla f\left(\mathbf{v}_{\tau,t}; z_{i_{\tau,t}^{(1)}}\right) - \nabla F_S\left(\mathbf{v}_{\tau,t}\right)\|^2\big] + \mathbb{E}\big[\|\nabla F_S\left(\mathbf{v}_{\tau,t}\right)\|^2\big]$$

$$= \frac{\mathbb{E}\big[\|\nabla f\left(\mathbf{v}_{\tau,t}; z_{i_{\tau,t}^{(1)}}\right)\|^2\big]}{b} - \frac{\mathbb{E}\big[\|\nabla F_S\left(\mathbf{v}_{\tau,t}\right)\|^2\big]}{b} + \mathbb{E}\big[\|\nabla F_S\left(\mathbf{v}_{\tau,t}\right)\|^2\big]$$

$$\leq \frac{2L\mathbb{E}\big[F_S(\mathbf{v}_{\tau,t})\big]}{b} + \mathbb{E}\big[\|\nabla F_S\left(\mathbf{v}_{\tau,t}\right)\|^2\big]$$

$$\leq \frac{2L\mathbb{E}\big[F_S(\mathbf{v}_{\tau,t})\big]}{b} + 2L\mathbb{E}[F_S(\mathbf{v}_{\tau,t}) - F_S(\mathbf{w}_S)], \tag{A.16}$$

where the last inequality follows from Lemma 10, where we set $y = \mathbf{w}_S$. We then analyze the single step in the inner-loop,

$$
\begin{aligned}
\mathbb{E}\left[\|\mathbf{v}_{\tau+1,t} - \mathbf{w}_S\|^2\right] &= \mathbb{E}\left[\|\mathbf{v}_{\tau,t} - \eta\nabla f(\mathbf{v}_{\tau,t}; B_{\tau,t}) - \mathbf{w}_S\|^2\right] \\
&= \mathbb{E}\left[\|\mathbf{v}_{\tau,t} - \mathbf{w}_S\|^2 - 2\eta\langle\mathbf{v}_{\tau,t} - \mathbf{w}_S, \nabla f(\mathbf{v}_{\tau,t}; B_{\tau,t})\rangle + \eta^2\|\nabla f(\mathbf{v}_{\tau,t}; B_{\tau,t})\|^2\right] \\
&= \mathbb{E}\left[\|\mathbf{v}_{\tau,t} - \mathbf{w}_S\|^2\right] - 2\eta\mathbb{E}\left[\langle\mathbf{v}_{\tau,t} - \mathbf{w}_S, \nabla F_S(\mathbf{v}_{\tau,t})\rangle\right] + \eta^2\mathbb{E}\left[\|\nabla f(\mathbf{v}_{\tau,t}; B_{\tau,t})\|^2\right].
\end{aligned}
$$
(A.17)

By convexity, we have $\langle\mathbf{v}_{\tau,t} - \mathbf{w}_S, \nabla F_S(\mathbf{v}_{\tau,t})\rangle \geq F_S(\mathbf{v}_{\tau,t}) - F_S(\mathbf{w}_S)$. Substituting this and the above result, we get

$$
\begin{aligned}
\mathbb{E}\left[\|\mathbf{v}_{\tau+1,t} - \mathbf{w}_S\|^2\right] &\leq \mathbb{E}\left[\|\mathbf{v}_{\tau,t} - \mathbf{w}_S\|^2\right] - 2\eta\mathbb{E}\left[F_S(\mathbf{v}_{\tau,t}) - F_S(\mathbf{w}_S)\right] + \eta^2\left(\frac{2L\mathbb{E}[F_S(\mathbf{v}_{\tau,t})]}{b} + 2L\mathbb{E}[F_S(\mathbf{v}_{\tau,t}) - F_S(\mathbf{w}_S)]\right) \\
&= \mathbb{E}\left[\|\mathbf{v}_{\tau,t} - \mathbf{w}_S\|^2\right] - \left(2\eta - \frac{2L\eta^2(b+1)}{b}\right)\mathbb{E}\left[F_S(\mathbf{v}_{\tau,t}) - F_S(\mathbf{w}_S)\right] + \frac{2L\eta^2\mathbb{E}[F_S(\mathbf{w}_S)]}{b}.
\end{aligned}
$$

It then follows that

$$
2\eta\left(1 - \frac{L\eta(b+1)}{b}\right)\mathbb{E}\left[F_S(\mathbf{v}_{\tau,t}) - F_S(\mathbf{w}_S)\right] \leq \mathbb{E}\left[\|\mathbf{v}_{\tau,t} - \mathbf{w}_S\|^2 - \|\mathbf{v}_{\tau+1,t} - \mathbf{w}_S\|^2\right] + \frac{2L\eta^2\mathbb{E}[F_S(\mathbf{w}_S)]}{b}.
$$

Recall the assumption of $\eta \leq \frac{b}{L(b+1)}$, we can divide by $2\eta(1 - \frac{L\eta(b+1)}{b})$ and get

$$
\mathbb{E}\left[F_S(\mathbf{v}_{\tau,t}) - F_S(\mathbf{w}_S)\right] \leq \frac{b}{2\eta(b - L\eta(b+1))}\mathbb{E}\left[\|\mathbf{v}_{\tau,t} - \mathbf{w}_S\|^2 - \|\mathbf{v}_{\tau+1,t} - \mathbf{w}_S\|^2\right] + \frac{L\eta\mathbb{E}[F_S(\mathbf{w}_S)]}{b - L\eta(b+1)}.
$$

We take an average of the above inequality from $\tau = 0$ to $k - 1$, and get

$$
\begin{aligned}
\frac{1}{k}\sum_{\tau=0}^{k-1}\mathbb{E}\left[F_S(\mathbf{v}_{\tau,t}) - F_S(\mathbf{w}_S)\right] &\leq \frac{b}{2\eta k(b - L\eta(b+1))}\sum_{\tau=0}^{k-1}\mathbb{E}\left[\|\mathbf{v}_{\tau,t} - \mathbf{w}_S\|^2 - \|\mathbf{v}_{\tau+1,t} - \mathbf{w}_S\|^2\right] + \frac{L\eta\mathbb{E}[F_S(\mathbf{w}_S)]}{b - L\eta(b+1)} \\
&= \frac{b}{2\eta k(b - L\eta(b+1))}\mathbb{E}\left[\|\mathbf{v}_{0,t} - \mathbf{w}_S\|^2 - \|\mathbf{v}_{k,t} - \mathbf{w}_S\|^2\right] + \frac{L\eta\mathbb{E}[F_S(\mathbf{w}_S)]}{b - L\eta(b+1)}.
\end{aligned}
$$
(A.18)

By the slow updating rule of Lookahead, we know $(1 - \alpha)(\mathbf{w}_{t-1} - \mathbf{w}^*) = (\mathbf{w}_t - \mathbf{w}^*) - \alpha(\mathbf{v}_{k,t} - \mathbf{w}^*)$ and get

$$
\|\mathbf{v}_{0,t} - \mathbf{w}_S\|^2 - \|\mathbf{v}_{k,t} - \mathbf{w}_S\|^2 = \|\mathbf{w}_{t-1} - \mathbf{w}_S\|^2 - \|\mathbf{v}_{k,t} - \mathbf{w}_S\|^2 \leq \frac{1}{\alpha}\left(\|\mathbf{w}_{t-1} - \mathbf{w}_S\|^2 - \|\mathbf{w}_t - \mathbf{w}_S\|^2\right).
$$

Substituting this into (A.18), we have

$$
\frac{1}{k}\sum_{\tau=0}^{k-1}\mathbb{E}\left[F_S(\mathbf{v}_{\tau,t}) - F_S(\mathbf{w}_S)\right] \leq \frac{b}{2\alpha\eta k(b - L\eta(b+1))}\mathbb{E}\left[\|\mathbf{w}_{t-1} - \mathbf{w}_S\|^2 - \|\mathbf{w}_t - \mathbf{w}_S\|^2\right] + \frac{L\eta\mathbb{E}[F_S(\mathbf{w}_S)]}{b - L\eta(b+1)}.
$$

We take an average of the above inequality and get

$$
\begin{aligned}
\frac{1}{kT}\sum_{t=1}^{T}\sum_{\tau=0}^{k-1}\mathbb{E}\left[F_S(\mathbf{v}_{\tau,t}) - F_S(\mathbf{w}_S)\right] &\leq \frac{b}{2\alpha\eta kT(b - L\eta(b+1))}\sum_{t=1}^{T}\mathbb{E}\left[\|\mathbf{w}_{t-1} - \mathbf{w}_S\|^2 - \|\mathbf{w}_t - \mathbf{w}_S\|^2\right] + \frac{L\eta\mathbb{E}[F_S(\mathbf{w}_S)]}{b - L\eta(b+1)} \\
&\leq \frac{b\mathbb{E}\left[\|\mathbf{w}_0 - \mathbf{w}_S\|^2\right]}{2\alpha\eta kT(b - L\eta(b+1))} + \frac{L\eta\mathbb{E}[F_S(\mathbf{w}_S)]}{b - L\eta(b+1)} \\
&\leq \frac{b\mathbb{E}\left[\|\mathbf{w}_0 - \mathbf{w}_S\|^2\right]}{2\alpha\eta kT(b - L\eta(b+1))} + \frac{L\eta\mathbb{E}[F_S(\mathbf{w}^*)]}{b - L\eta(b+1)}.
\end{aligned}
$$
(A.19)

We complete the proof by applying the Jensen's inequality. $\qquad\square$

*Proof of Theorem 3.* By Lemma 1 (part (b)) and (5.2), we have (note our stability bounds also apply to $\bar{\mathbf{v}}_R$ due to the convexity of norm)

$$
\mathbb{E}\left[F(\bar{\mathbf{v}}_R) - F_S(\bar{\mathbf{v}}_R)\right] \leq \frac{L}{\gamma}\mathbb{E}[F_S(\bar{\mathbf{v}}_R)] + (L + \gamma)\left(\frac{8\alpha^2 L}{nb} + \frac{8\alpha^2 LTk}{n^2}\right)\sum_{h=1}^{T}\sum_{j=0}^{k-1}\eta_{j,h}^2\mathbb{E}\left[F_S(\mathbf{v}_{j,h})\right].
$$
(A.20)

By (A.19) we know that

$$\frac{1}{kT} \sum_{t=1}^{T} \sum_{\tau=0}^{k-1} \mathbb{E}\left[F_S\left(\mathbf{v}_{\tau,t}\right)\right] \lesssim F(\mathbf{w}^*) + \frac{L\eta F(\mathbf{w}^*)}{b} + \frac{1}{\alpha\eta kT}. \tag{A.21}$$

Let $R = Tk$. We combine the above inequalities and get

$$\mathbb{E}\left[F(\overline{\mathbf{v}}_R) - F_S(\overline{\mathbf{v}}_R)\right] \lesssim \frac{L(F(\mathbf{w}^*) + L\eta F(\mathbf{w}^*)/b + 1/(\alpha\eta R))}{\gamma}$$

$$+ L(L+\gamma)\alpha^2\eta^2\left(\frac{1}{nb} + \frac{R}{n^2}\right)(RF(\mathbf{w}^*) + RL\eta F(\mathbf{w}^*)/b + 1/(\alpha\eta)). \tag{A.22}$$

We plug (A.22) and the optimization error bound (A.15) back into (3.1) and get

$$\mathbb{E}\left[F(\overline{\mathbf{v}}_R)\right] - F(\mathbf{w}^*) \lesssim \frac{L\eta F(\mathbf{w}^*)}{b} + \frac{1}{\alpha\eta R} + \frac{F(\mathbf{w}^*) + L\eta F(\mathbf{w}^*)/b + 1/(\alpha\eta R)}{\gamma} +$$

$$L(L+\gamma)\alpha^2\eta^2\left(\frac{1}{nb} + \frac{R}{n^2}\right)(RF(\mathbf{w}^*) + RL\eta F(\mathbf{w}^*)/b + 1/(\alpha\eta)).$$

The proof is completed. $\qquad\square$

*Proof of Corollary 4.* We first consider the case $F(\mathbf{w}^*) \geq \frac{1}{n}$. Fix any constant $\alpha \in (0,1]$, we choose $\eta = \frac{b}{\sqrt{nF(\mathbf{w}^*)}}$, $R \asymp \frac{n}{b}$, and $\gamma = \sqrt{nF(\mathbf{w}^*)} \geq 1$. Note the assumption $b \leq \sqrt{nF(\mathbf{w}^*)}/(2L)$ ensures that $\eta \leq 1/(2L)$. Then Eq. (5.3) implies

$$\mathbb{E}\left[F(\overline{\mathbf{v}}_R) - F(\mathbf{w}^*)\right] \lesssim \frac{LF(\mathbf{w}^*)}{\sqrt{nF(\mathbf{w}^*)}} + \frac{F(\mathbf{w}^*)^{\frac{1}{2}}}{\sqrt{n}} + \frac{(nF(\mathbf{w}^*))^{\frac{1}{2}} + L + 1}{n}$$

$$+ \frac{2L}{n^2 F(\mathbf{w}^*)}\left(L + (nF(\mathbf{w}^*))^{\frac{1}{2}})\right)\left(nF(\mathbf{w}^*) + (L+1)(nF(\mathbf{w}^*))^{\frac{1}{2}}\right)$$

$$\lesssim \frac{LF(\mathbf{w}^*)^{1/2}}{\sqrt{n}} + \frac{L^2}{n}.$$

We now consider the case $F(\mathbf{w}^*) < \frac{1}{n}$. We fix $\alpha \in (0,1]$ as a constant, and choose $\eta = \frac{1}{2L}$, $R \asymp n$, and $\gamma = 1$. Then Eq. (5.3) implies

$$\mathbb{E}\left[F(\overline{\mathbf{v}}_R) - F(\mathbf{w}^*)\right] \lesssim F(\mathbf{w}^*) + \frac{L}{n} + \frac{L+1}{4nL}\left(nF(\mathbf{w}^*) + 2L\right) \lesssim \frac{L}{n} + F(\mathbf{w}^*).$$

The proof is completed. $\qquad\square$

## A.3 PROOF OF THEOREM 5

*Proof.* Recalling from Eq. (A.1) the refined Lookahead updating rule, we have

$$\|\mathbf{w}_{t+1} - \mathbf{w}_{t+1}^{(i)}\|_2$$

$$\leq (1-\alpha)\|\mathbf{w}_t - \mathbf{w}_t^{(i)}\|_2 + \alpha\Big\|\mathbf{v}_{k-1,t+1} - \frac{\eta_{k-1,t+1}}{b}\sum_{m:m\neq i}^{n} A_{k-1,t+1}^{(m)}\nabla f(\mathbf{v}_{k-1,t+1}; z_m) - \mathbf{v}_{k-1,t+1}^{(i)}$$

$$+ \frac{\eta_{k-1,t+1}}{b}\sum_{m:m\neq i}^{n} A_{k-1,t+1}^{(m)}\nabla f(\mathbf{v}_{k-1,t+1}^{(i)}; z_m)\Big\|_2 + \frac{\alpha A_{k-1,t+1}^{(i)}\eta_{k-1,t+1}}{b}\|\nabla f\left(\mathbf{v}_{k-1,t+1}; z_i\right) - \nabla f(\mathbf{v}_{k-1,t+1}^{(i)}; z_i')\|_2.$$

Since $f$ is smooth and $\sum_{m:m\neq i}^{n} A_{k-1,t+1}^{(m)} \leq b$, therefore $\mathbf{v} \mapsto \frac{1}{b}\sum_{m:m\neq i}^{n} A_{k-1,t+1}^{(m)} f(\mathbf{v}; z_m)$ is $L$-smooth. It follows from Lemma 8 and the assumption $\eta_{k-1,t+1} \leq \frac{1}{L}$ that

$$\|\mathbf{w}_{t+1} - \mathbf{w}_{t+1}^{(i)}\|_2 \leq (1-\alpha)\|\mathbf{w}_t - \mathbf{w}_t^{(i)}\|_2 +$$

$$\frac{\alpha\eta_{k-1,t+1}A_{k-1,t+1}^{(i)}\mathfrak{C}_{k-1,t+1}^{(i)}}{b} + \alpha\left(1 - \frac{\mu\eta_{k-1,t+1}}{2}\right)\|\mathbf{v}_{k-1,t+1} - \mathbf{v}_{k-1,t+1}^{(i)}\|_2. \tag{A.23}$$

We take the expectation on both sides and get

$$\mathbb{E}\big[\|\mathbf{w}_{t+1} - \mathbf{w}_{t+1}^{(i)}\|_2\big] \le (1-\alpha)\,\mathbb{E}\big[\|\mathbf{w}_t - \mathbf{w}_t^{(i)}\|_2\big] + \frac{2\alpha\eta_{k-1,t+1}\sqrt{2L\mathbb{E}\left[f\left(\mathbf{v}_{k-1,t+1}; z_i\right)\right]}}{n}$$
$$+ \alpha\left(1 - \frac{\mu\eta_{k-1,t+1}}{2}\right)\mathbb{E}\big[\|\mathbf{v}_{k-1,t+1} - \mathbf{v}_{k-1,t+1}^{(i)}\|_2\big],$$

where we have used (A.4) and (A.6). We do the iteration on inner-loop, and get

$$\mathbb{E}\big[\|\mathbf{w}_{t+1} - \mathbf{w}_{t+1}^{(i)}\|_2\big] \le (1-\alpha)\,\mathbb{E}\big[\|\mathbf{w}_t - \mathbf{w}_t^{(i)}\|_2\big] + \frac{2\alpha\sqrt{2L}}{n}\sum_{j=0}^{k-1}\eta_{j,t+1}\sqrt{\mathbb{E}\left[f\left(\mathbf{v}_{j,t+1}; z_i\right)\right]}\prod_{j'=j+1}^{k-1}\left(1 - \frac{\mu\eta_{j',t+1}}{2}\right)$$

$$+ \alpha\mathbb{E}\big[\|\mathbf{w}_t - \mathbf{w}_t^{(i)}\|_2\big]\prod_{j=0}^{k-1}\left(1 - \frac{\mu\eta_{j,t+1}}{2}\right)$$

$$\le (1 - \frac{\alpha}{2})\mathbb{E}\big[\|\mathbf{w}_t - \mathbf{w}_t^{(i)}\|_2\big] + \frac{2\alpha\sqrt{2L}}{n}\sum_{j=0}^{k-1}\eta_{j,t+1}\sqrt{\mathbb{E}\left[f\left(\mathbf{v}_{j,t+1}; z_i\right)\right]}\prod_{j'=j+1}^{k-1}\left(1 - \frac{\mu\eta_{j',t+1}}{2}\right),$$

where we have used the following inequality due to the the assumption $\eta_{j,t+1} \ge \frac{2\ln 2}{k\mu}$

$$\prod_{j=0}^{k-1}\left(1 - \frac{\mu\eta_{j,t+1}}{2}\right) \le \exp\left(-\sum_{j=0}^{k}\frac{\mu\eta_{j,t+1}}{2}\right) \le \exp\left(-k\frac{\mu 2\log 2}{2k\mu}\right) = \frac{1}{2}. \qquad \text{(A.24)}$$

By iteration on outer-loop,

$$\mathbb{E}\big[\|\mathbf{w}_{t+1} - \mathbf{w}_{t+1}^{(i)}\|_2\big] \le \frac{2\alpha\sqrt{2L}}{n}\sum_{t'=1}^{t+1}(1 - \frac{\alpha}{2})^{t+1-t'}\sum_{j=0}^{k-1}\eta_{j,t'}\sqrt{\mathbb{E}\left[f\left(\mathbf{v}_{j,t'}; z_i\right)\right]}\prod_{j'=j+1}^{k-1}\left(1 - \frac{\mu\eta_{j',t'}}{2}\right).$$
$$\text{(A.25)}$$

Taking an average over $i$ and using the concavity of $x \mapsto \sqrt{x}$, we get

$$\frac{1}{n}\sum_{i=1}^{n}\mathbb{E}\big[\|\mathbf{w}_{t+1} - \mathbf{w}_{t+1}^{(i)}\|_2\big] \le \frac{2\alpha\sqrt{2L}}{n^2}\sum_{t'=1}^{t+1}(1 - \frac{\alpha}{2})^{t+1-t'}\sum_{j=0}^{k-1}\sum_{i=1}^{n}\eta_{j,t'}\sqrt{\mathbb{E}\left[f\left(\mathbf{v}_{j,t'}; z_i\right)\right]}\prod_{j'=j+1}^{k-1}\left(1 - \frac{\mu\eta_{j',t'}}{2}\right)$$

$$\le \frac{2\alpha\sqrt{2L}}{n}\sum_{t'=1}^{t+1}(1 - \frac{\alpha}{2})^{t+1-t'}\sum_{j=0}^{k-1}\eta_{j,t'}\Big(\frac{1}{n}\sum_{i=1}^{n}\mathbb{E}\left[f\left(\mathbf{v}_{j,t'}; z_i\right)\right]\Big)^{\frac{1}{2}}\prod_{j'=j+1}^{k-1}\left(1 - \frac{\mu\eta_{j',t'}}{2}\right)$$

$$= \frac{2\alpha\sqrt{2L}}{n}\sum_{t'=1}^{t+1}(1 - \frac{\alpha}{2})^{t+1-t'}\sum_{j=0}^{k-1}\eta_{j,t'}\sqrt{\mathbb{E}\left[F_S\left(\mathbf{v}_{j,t'}\right)\right]}\prod_{j'=j+1}^{k-1}\left(1 - \frac{\mu\eta_{j',t'}}{2}\right).$$

This established the stated $\ell_1$-stability bound (5.4).
We now prove Eq. (5.5). Recall Eq. (A.2), we do iteration on inner-loop in Eq. (A.23) and get

$$\|\mathbf{w}_{t+1} - \mathbf{w}_{t+1}^{(i)}\|_2$$

$$\le (1-\alpha)\|\mathbf{w}_t - \mathbf{w}_t^{(i)}\|_2 + \frac{\alpha}{b}\sum_{j=0}^{k-1}\eta_{j,t+1}A_{j,t+1}^{(i)}\mathfrak{C}_{j,t+1}^{(i)}\prod_{j'=j+1}^{k-1}\left(1 - \frac{\mu\eta_{j',t+1}}{2}\right) + \alpha\|\mathbf{w}_t - \mathbf{w}_t^{(i)}\|_2\prod_{j=0}^{k-1}\left(1 - \frac{\mu\eta_{j,t+1}}{2}\right)$$

$$\le \|\mathbf{w}_t - \mathbf{w}_t^{(i)}\|_2 + \frac{\alpha}{b}\sum_{j=0}^{k-1}\eta_{j,t+1}A_{j,t+1}^{(i)}\mathfrak{C}_{j,t+1}^{(i)}\prod_{j'=j+1}^{k-1}\left(1 - \frac{\mu\eta_{j',t+1}}{2}\right).$$

Then we iterate on outer-loop and get

$$\|\mathbf{w}_{t+1} - \mathbf{w}_{t+1}^{(i)}\|_2 \le \frac{\alpha}{b}\sum_{t'=1}^{t+1}\sum_{j=0}^{k-1}\eta_{j,t'}A_{j,t'}^{(i)}\mathfrak{C}_{j,t'}^{(i)}\prod_{j'=j+1}^{k-1}\left(1 - \frac{\mu\eta_{j',t'}}{2}\right)$$

$$= \frac{\alpha}{b}\sum_{t'=1}^{t+1}\sum_{j=0}^{k-1}\eta_{j,t'}\Big(A_{j,t'}^{(i)} - \frac{b}{n}\Big)\mathfrak{C}_{j,t'}^{(i)}\prod_{j'=j+1}^{k-1}\left(1 - \frac{\mu\eta_{j',t'}}{2}\right) + \frac{\alpha}{n}\sum_{t'=1}^{t+1}\sum_{j=0}^{k-1}\eta_{j,t'}\mathfrak{C}_{j,t'}^{(i)}\prod_{j'=j+1}^{k-1}\left(1 - \frac{\mu\eta_{j',t'}}{2}\right).$$

By taking the square and the expectation on both sides, we get

$$
\mathbb{E}\big[\|\mathbf{w}_{t+1} - \mathbf{w}_{t+1}^{(i)}\|_2^2\big]
$$

$$
\leq \frac{2\alpha^2}{b^2}\mathbb{E}\Big[\Big(\sum_{t'=1}^{t+1}\sum_{j=0}^{k-1}\eta_{j,t'}\big(A_{j,t'}^{(i)} - \frac{b}{n}\big)\mathfrak{C}_{j,t'}^{(i)}\prod_{j'=j+1}^{k-1}\big(1-\frac{\mu\eta_{j',t'}}{2}\big)\Big)^2\Big] + \frac{2\alpha^2}{n^2}\mathbb{E}\Big[\Big(\sum_{t'=1}^{t+1}\sum_{j=0}^{k-1}\eta_{j,t'}\mathfrak{C}_{j,t'}^{(i)}\prod_{j'=j+1}^{k-1}\big(1-\frac{\mu\eta_{j',t'}}{2}\big)\Big)^2\Big]
$$

$$
= \frac{2\alpha^2}{b^2}\sum_{t'=1}^{t+1}\sum_{j=0}^{k-1}\eta_{j,t'}^2\,\mathbb{E}\Big[\big(A_{j,t'}^{(i)} - \frac{b}{n}\big)^2(\mathfrak{C}_{j,t'}^{(i)})^2\prod_{j'=j+1}^{k-1}\big(1-\frac{\mu\eta_{j',t'}}{2}\big)^2\Big] + \frac{2\alpha^2}{n^2}\mathbb{E}\Big[\Big(\sum_{t'=1}^{t+1}\sum_{j=0}^{k-1}\eta_{j,t'}\mathfrak{C}_{j,t'}^{(i)}\prod_{j'=j+1}^{k-1}\big(1-\frac{\mu\eta_{j',t'}}{2}\big)\Big)^2\Big]
$$

$$
\leq \frac{2\alpha^2}{nb}\sum_{t'=1}^{t+1}\sum_{j=0}^{k-1}\eta_{j,t'}^2\,\mathbb{E}\Big[(\mathfrak{C}_{j,t'}^{(i)})^2\Big]\prod_{j'=j+1}^{k-1}\big(1-\frac{\mu\eta_{j',t'}}{2}\big)^2 + \frac{2\alpha^2}{n^2}\mathbb{E}\Big[\Big(\sum_{t'=1}^{t+1}\sum_{j=0}^{k-1}\eta_{j,t'}\mathfrak{C}_{j,t'}^{(i)}\prod_{j'=j+1}^{k-1}\big(1-\frac{\mu\eta_{j',t'}}{2}\big)\Big)^2\Big]
$$

$$
\tag{A.26}
$$

where we used (A.11) and $\mathbb{E}_{B_{j,t'}}\big[(A_{j,t'}^{(i)} - \frac{b}{n})^2\big] \leq \frac{b}{n}$. For the second term, we apply the Cauchy-Schwarz inequality,

$$
\Big(\sum_{t'=1}^{t+1}\sum_{j=0}^{k-1}\eta_{j,t'}\mathfrak{C}_{j,t'}^{(i)}\prod_{j'=j+1}^{k-1}\big(1-\frac{\mu\eta_{j',t'}}{2}\big)\Big)^2
$$

$$
\leq \Big(\sum_{t'=1}^{t+1}\sum_{j=0}^{k-1}\eta_{j,t'}(\mathfrak{C}_{j,t'}^{(i)})^2\prod_{j'=j+1}^{k-1}\big(1-\frac{\mu\eta_{j',t'}}{2}\big)\Big)\Big(\sum_{t'=1}^{t+1}\sum_{j=0}^{k-1}\eta_{j,t'}\prod_{j'=j+1}^{k-1}\big(1-\frac{\mu\eta_{j',t'}}{2}\big)\Big)
$$

$$
\leq \frac{2(t+1)}{\mu}\Big(\sum_{t'=1}^{t+1}\sum_{j=0}^{k-1}\eta_{j,t'}(\mathfrak{C}_{j,t'}^{(i)})^2\prod_{j'=j+1}^{k-1}\big(1-\frac{\mu\eta_{j',t'}}{2}\big)\Big),
\tag{A.27}
$$

where the following result is used in the last inequality

$$
\sum_{j=0}^{k-1}\eta_{j,t'}\prod_{j'=j+1}^{k-1}\big(1-\frac{\mu\eta_{j',t'}}{2}\big) = \frac{2}{\mu}\sum_{j=0}^{k-1}\big(1-\big(1-\frac{\mu\eta_{j,t'}}{2}\big)\big)\prod_{j'=j+1}^{k-1}\big(1-\frac{\mu\eta_{j',t'}}{2}\big)
$$

$$
= \frac{2}{\mu}\sum_{j=0}^{k-1}\Big(\prod_{j'=j+1}^{k-1}\big(1-\frac{\mu\eta_{j',t'}}{2}\big) - \prod_{j'=j}^{k-1}\big(1-\frac{\mu\eta_{j',t'}}{2}\big)\Big)
$$

$$
= \frac{2}{\mu}\Big(1-\prod_{j'=0}^{k-1}\big(1-\frac{\mu\eta_{j',t'}}{2}\big)\Big) \leq \frac{2}{\mu}.
\tag{A.28}
$$

Combining the above discussions together, we further get

$$
\mathbb{E}\big[\|\mathbf{w}_{t+1} - \mathbf{w}_{t+1}^{(i)}\|_2^2\big] \leq \sum_{t'=1}^{t+1}\sum_{j=0}^{k-1}\Big(\frac{2\alpha^2\eta_{j,t'}^2}{nb} + \frac{4(t+1)\alpha^2\eta_{j,t'}}{n^2\mu}\Big)\mathbb{E}\big[(\mathfrak{C}_{j,t'}^{(i)})^2\big]\prod_{j'=j+1}^{k-1}\big(1-\frac{\mu\eta_{j',t'}}{2}\big).
$$

Recalling result in (A.12), $\mathbb{E}\big[(\mathfrak{C}_{j,t'}^{(i)})^2\big] \leq 8L\mathbb{E}\left[f\left(\mathbf{v}_{j,h}; z_i\right)\right]$, we further derive

$$
\mathbb{E}\big[\|\mathbf{w}_{t+1} - \mathbf{w}_{t+1}^{(i)}\|_2^2\big] \leq \sum_{t'=1}^{t+1}\sum_{j=0}^{k-1}\Big(\frac{16\alpha^2\eta_{j,t'}^2}{nb} + \frac{32(t+1)\alpha^2\eta_{j,t'}}{n^2\mu}\Big)\mathbb{E}\left[f\left(\mathbf{v}_{j,t'}; z_i\right)\right]\prod_{j'=j+1}^{k-1}\big(1-\frac{\mu\eta_{j',t'}}{2}\big).
$$

Taking an average over $i \in [n]$, we get the stated bound

$$
\frac{1}{n}\sum_{i=1}^{n}\mathbb{E}\big[\|\mathbf{w}_{t+1} - \mathbf{w}_{t+1}^{(i)}\|_2^2\big] \leq \sum_{t'=1}^{t+1}\sum_{j=0}^{k-1}\Big(\frac{16\alpha^2\eta_{j,t'}^2}{nb} + \frac{32(t+1)\alpha^2\eta_{j,t'}}{n^2\mu}\Big)\mathbb{E}\left[F_S\left(\mathbf{v}_{j,t'}\right)\right]\prod_{j'=j+1}^{k-1}\big(1-\frac{\mu\eta_{j',t'}}{2}\big).
$$

The proof is completed. $\qquad\square$

## A.4 PROOF OF THEOREM 6

We first state and prove the optimization error bound.

**Lemma 11** (Optimization Error of Lookahead: Strongly Convex Case). *Suppose the assumptions in Theorem 5 hold, by setting the learning rate $\eta = \frac{b\mu}{2L^2(b+1)}$, the optimization error of the output $\mathbf{w}_T$ of Lookahead satisfies*

$$\mathbb{E}[F_S(\mathbf{w}_T) - F_S(\mathbf{w}^*)] \leq \frac{L}{2}\, e^{-\frac{3}{4}\alpha k\mu\eta T}\, \mathbb{E}\big[\|\mathbf{w}_0 - \mathbf{w}_S\|^2\big]$$

$$+ \frac{L\alpha}{2} \sum_{t=0}^{T-1} e^{-\frac{3}{4}\alpha k\mu\eta t} \sum_{k'=0}^{k-1} e^{-\frac{3}{4}\mu\eta k'}\, \frac{2\eta^2 L}{b}\, \mathbb{E}[F_S(\mathbf{w}_S)]. \qquad (A.29)$$

*Furthermore, by choosing $b \lesssim n$, $k = \frac{2L}{\alpha\mu}$, and $T \asymp \log(\mu n)$, we have*

$$\mathbb{E}[F_S(\mathbf{w}_T) - F_S(\mathbf{w}^*)] \lesssim \frac{L}{n}\mathbb{E}[\|\mathbf{w}_0 - \mathbf{w}_S\|^2] + \mathbb{E}[F_S(\mathbf{w}_S)]. \qquad (A.30)$$

*Proof.* Since $F_S(\mathbf{w}_S) \leq F_S(\mathbf{w}^*)$, an upper bound for $F_S(\mathbf{w}_T) - F_S(\mathbf{w}_S)$ is also an upper bound for $F_S(\mathbf{w}_T) - F_S(\mathbf{w}^*)$. Since the function $F_S(\mathbf{w})$ is $\mu$-strongly convex and $\mathbf{w}_S$ is the optimum of $F_S(\mathbf{w})$, we have

$$F_S(\mathbf{v}_{\tau-1,t}) \geq F_S(\mathbf{w}_S) + \langle \nabla F_S(\mathbf{w}_S),\, \mathbf{w}_S - \mathbf{v}_{\tau-1,t} \rangle + \frac{\mu}{2}\|\mathbf{w}_S - \mathbf{v}_{\tau-1,t}\|_2^2$$

$$= F_S(\mathbf{w}_S) + \frac{\mu}{2}\|\mathbf{w}_S - \mathbf{v}_{\tau-1,t}\|_2^2.$$

Similarly, we have

$$F_S(\mathbf{w}_S) \geq F_S(\mathbf{v}_{\tau-1,t}) + \langle \nabla F_S(\mathbf{v}_{\tau-1,t}),\, \mathbf{w}_S - \mathbf{v}_{\tau-1,t} \rangle + \frac{\mu}{2}\|\mathbf{w}_S - \mathbf{v}_{\tau-1,t}\|_2^2.$$

It then follows that

$$\mathbb{E}[\|\mathbf{v}_{\tau,t} - \mathbf{w}_S\|^2] = \mathbb{E}[\|\mathbf{v}_{\tau-1,t} - \eta\,\nabla f(\mathbf{v}_{\tau-1,t}; B_{\tau-1,t}) - \mathbf{w}_S\|^2]$$

$$= \mathbb{E}[\|\mathbf{v}_{\tau-1,t} - \mathbf{w}_S\|^2 - 2\eta\langle \mathbf{v}_{\tau-1,t} - \mathbf{w}_S, \nabla f(\mathbf{v}_{\tau-1,t}; B_{\tau-1,t})\rangle + \eta^2\|\nabla f(\mathbf{v}_{\tau-1,t}; B_{\tau-1,t})\|^2]$$

$$= \mathbb{E}[\|\mathbf{v}_{\tau-1,t} - \mathbf{w}_S\|^2 - 2\eta\langle \mathbf{v}_{\tau-1,t} - \mathbf{w}_S, \nabla F_S(\mathbf{v}_{\tau-1,t})\rangle + \eta^2\|\nabla f(\mathbf{v}_{\tau-1,t}; B_{\tau-1,t})\|^2]$$

$$\leq \mathbb{E}[\|\mathbf{v}_{\tau-1,t} - \mathbf{w}_S\|^2 + 2\eta\big(F_S(\mathbf{w}_S) - F_S(\mathbf{v}_{\tau-1,t}) - \frac{\mu}{2}\|\mathbf{w}_S - \mathbf{v}_{\tau-1,t}\|_2^2\big) + \eta^2\|\nabla f(\mathbf{v}_{\tau-1,t}; B_{\tau-1,t})\|^2]$$

$$\leq \mathbb{E}[\|\mathbf{v}_{\tau-1,t} - \mathbf{w}_S\|^2 + 2\eta\big(-\frac{\mu}{2}\|\mathbf{w}_S - \mathbf{v}_{\tau-1,t}\|_2^2 - \frac{\mu}{2}\|\mathbf{w}_S - \mathbf{v}_{\tau-1,t}\|_2^2\big) + \eta^2\|\nabla f(\mathbf{v}_{\tau-1,t}; B_{\tau-1,t})\|^2]$$

$$\leq (1 - 2\mu\eta)\mathbb{E}[\|\mathbf{v}_{\tau-1,t} - \mathbf{w}_S\|^2] + \eta^2\mathbb{E}[\|\nabla f(\mathbf{v}_{\tau-1,t}; B_{\tau-1,t})\|^2].$$

For the second term, we use the result of (A.16) and have

$$\mathbb{E}[\|\mathbf{v}_{\tau,t} - \mathbf{w}_S\|^2] \leq (1 - 2\mu\eta)\mathbb{E}[\|\mathbf{v}_{\tau-1,t} - \mathbf{w}_S\|^2] + \eta^2\frac{2L\mathbb{E}[F_S(\mathbf{v}_{\tau-1,t})]}{b} + 2L\eta^2\mathbb{E}[F_S(\mathbf{v}_{\tau-1,t}) - F_S(\mathbf{w}_S)]$$

$$\leq (1 - 2\mu\eta + \eta^2 L^2)\mathbb{E}[\|\mathbf{v}_{\tau-1,t} - \mathbf{w}_S\|^2] + \eta^2\frac{2L\mathbb{E}[F_S(\mathbf{v}_{\tau-1,t}) - F_S(\mathbf{w}_S)] + 2L\mathbb{E}[F_S(\mathbf{w}_S)]}{b}$$

$$\leq (1 - 2\mu\eta + \eta^2\frac{L^2(b+1)}{b})\mathbb{E}[\|\mathbf{v}_{\tau-1,t} - \mathbf{w}_S\|^2] + \eta^2\frac{2L\mathbb{E}[F_S(\mathbf{w}_S)]}{b},$$

where we have used $F_S(\mathbf{w}) - F_S(\mathbf{w}_S) \leq \frac{L}{2}\|\mathbf{w} - \mathbf{w}_S\|_2^2$. For simplicity, we define $C$ as

$$C = \frac{L^2(b+1)}{b}.$$

The recurrence relation simplifies as

$$\mathbb{E}[\|\mathbf{v}_{\tau,t} - \mathbf{w}_S\|^2] \leq \big(1 - 2\mu\eta + C\eta^2\big)\,\mathbb{E}[\|\mathbf{v}_{\tau-1,t} - \mathbf{w}_S\|^2] + \eta^2\frac{2L\mathbb{E}[F_S(\mathbf{w}_S)]}{b}. \qquad (A.31)$$

We now choose

$$\eta = \frac{\mu}{2C} = \frac{\mu b}{2L^2(b+1)}.$$

Substituting this value back into the multiplicative factor gives

$$1 - 2\mu\left(\frac{\mu}{2C}\right) + C\left(\frac{\mu}{2C}\right)^2 = 1 - \frac{\mu^2}{C} + \frac{\mu^2}{4C} = 1 - \frac{3\mu^2}{4C} = 1 - \frac{3}{2}\mu\eta.$$

With this choice, the one-step recurrence (A.31) becomes

$$\mathbb{E}[\|\mathbf{v}_{\tau,t} - \mathbf{w}_S\|^2] \leq \left(1 - \frac{3}{2}\mu\eta\right)\mathbb{E}[\|\mathbf{v}_{\tau-1,t} - \mathbf{w}_S\|^2] + \eta^2\frac{2L\mathbb{E}[F_S(\mathbf{w}_S)]}{b}.$$

By applying the previous inequality recursively for the inner loop, we have

$$\mathbb{E}[\|\mathbf{v}_{k,t} - \mathbf{w}_S\|^2] \leq \left(1 - \frac{3}{2}\mu\eta\right)^k \mathbb{E}[\|\mathbf{w}_{t-1} - \mathbf{w}_S\|^2] + \sum_{k'=0}^{k-1} \left(1 - \frac{3}{2}\mu\eta\right)^{k'} \eta^2 \frac{2L\mathbb{E}[F_S(\mathbf{w}_S)]}{b}.$$

We now substitute this result back to the outer-loop. Recall the slow weights recurrence $\mathbf{w}_t = (1 - \alpha)\mathbf{w}_{t-1} + \alpha\mathbf{v}_{k,t}$,

$$\|\mathbf{w}_t - \mathbf{w}_S\|^2 = \|(1 - \alpha)(\mathbf{w}_{t-1} - \mathbf{w}_S) + \alpha(\mathbf{v}_{k,t} - \mathbf{w}_S)\|^2$$
$$\leq (1 - \alpha)\|\mathbf{w}_{t-1} - \mathbf{w}_S\|^2 + \alpha\|\mathbf{v}_{k,t} - \mathbf{w}_S\|^2.$$

Taking the expectation gives

$$\mathbb{E}[\|\mathbf{w}_t - \mathbf{w}_S\|^2] \leq (1 - \alpha)\mathbb{E}[\|\mathbf{w}_{t-1} - \mathbf{w}_S\|^2] + \alpha\mathbb{E}[\|\mathbf{v}_{k,t} - \mathbf{w}_S\|^2]$$

$$\leq (1 - \alpha)\mathbb{E}[\|\mathbf{w}_{t-1} - \mathbf{w}_S\|^2] + \alpha\left(1 - \frac{3}{2}\mu\eta\right)^k \mathbb{E}[\|\mathbf{w}_{t-1} - \mathbf{w}_S\|^2] + \alpha\sum_{k'=0}^{k-1}\left(1 - \frac{3}{2}\mu\eta\right)^{k'}\frac{2L\eta^2\mathbb{E}[F_S(\mathbf{w}_S)]}{b}$$

$$= \left[1 - \alpha + \alpha\left(1 - \frac{3}{2}\mu\eta\right)^k\right]\mathbb{E}[\|\mathbf{w}_{t-1} - \mathbf{w}_S\|^2] + \alpha\sum_{k'=0}^{k-1}\left(1 - \frac{3}{2}\mu\eta\right)^{k'}\frac{2L\eta^2\mathbb{E}[F_S(\mathbf{w}_S)]}{b}$$

$$= \left[1 - \alpha\left(1 - \left(1 - \frac{3}{2}\mu\eta\right)^k\right)\right]\mathbb{E}[\|\mathbf{w}_{t-1} - \mathbf{w}_S\|^2] + \alpha\sum_{k'=0}^{k-1}\left(1 - \frac{3}{2}\mu\eta\right)^{k'}\frac{2L\eta^2\mathbb{E}[F_S(\mathbf{w}_S)]}{b}.$$

Let $\rho$ be the contraction factor for the outer loop:

$$\rho = 1 - \alpha\left(1 - \left(1 - \frac{3}{2}\mu\eta\right)^k\right).$$

Since $0 < (1 - \frac{3}{2}\mu^2/C) < 1$ and $\alpha > 0$, we have $0 < \rho < 1$. Unwinding this recurrence from $t = 1$ to $T$:

$$\mathbb{E}[\|\mathbf{w}_t - \mathbf{w}_S\|^2] \leq \rho^t\mathbb{E}[\|\mathbf{w}_0 - \mathbf{w}_S\|^2] + \alpha\sum_{t'=0}^{t-1}\rho^{t'}\sum_{k'=0}^{k-1}\left(1 - \frac{3}{2}\mu\eta\right)^{k'}\frac{2L\eta^2\mathbb{E}[F_S(\mathbf{w}_S)]}{b}. \qquad \text{(A.32)}$$

Finally, using the $L$-smoothness property, $\mathbb{E}[F_S(\mathbf{w}_t) - F_S(\mathbf{w}_S)] \leq \frac{L}{2}\mathbb{E}[\|\mathbf{w}_t - \mathbf{w}_S\|^2]$, we arrive at the final optimization error bound.

$$\mathbb{E}[F_S(\mathbf{w}_T) - F_S(\mathbf{w}_S)] \leq \frac{L}{2}\left[1 - \alpha\left(1 - \left(1 - \frac{3}{2}\mu\eta\right)^k\right)\right]^T \mathbb{E}[\|\mathbf{w}_0 - \mathbf{w}_S\|^2]$$

$$+ \frac{L\alpha}{2}\sum_{t'=0}^{T-1}\left[1 - \alpha\left(1 - \left(1 - \frac{3}{2}\mu\eta\right)^k\right)\right]^{t'}\sum_{k'=0}^{k-1}\left(1 - \frac{3}{2}\mu\eta\right)^{k'}\frac{2L\eta^2\mathbb{E}[F_S(\mathbf{w}_S)]}{b}. \qquad \text{(A.33)}$$

We use the inequalities $1 + x \leq e^x$ for all real $x$ and $1 - e^{-x} \geq \frac{x}{1+x}$ for all $x \geq 0$ to get the following.

$$\left[1 - \alpha\left(1 - (1 - \frac{3}{2}\mu\eta)^k\right)\right]^T \leq \exp\left\{-\alpha\left[1 - (1 - \frac{3}{2}\mu\eta)^k\right]T\right\}$$

$$\leq \exp\left\{-\alpha\left(1 - \exp\left\{-\frac{3}{2}k\mu\eta\right\}\right)T\right\}$$

$$\leq \exp\left\{-\alpha\frac{3k\mu\eta}{3k\mu\eta + 2}T\right\}.$$

Then the optimization error bound becomes

$$\mathbb{E}[F_S(\mathbf{w}_T) - F_S(\mathbf{w}_S)] \leq \frac{L}{2}\exp\left\{-\alpha\frac{3k\mu\eta}{3k\mu\eta + 2}T\right\}\mathbb{E}\left[\|\mathbf{w}_0 - \mathbf{w}_S\|^2\right]$$

$$+ \frac{L\alpha}{2}\sum_{t=0}^{T-1}\exp\left\{-\alpha\frac{3k\mu\eta}{3k\mu\eta + 2}t\right\}\sum_{k'=0}^{k-1}\exp\left\{-\frac{3}{2}\mu\eta k'\right\}\frac{2\eta^2 L}{b}\mathbb{E}[F_S(\mathbf{w}_S)]. \quad \text{(A.34)}$$

We now choose the parameters to be $k = \frac{2L}{\mu\alpha}$, $T \asymp \log(n)$, and we fix $\alpha$. Since $b \geq 1$, we have $L\eta = \frac{b\mu}{2L(b+1)} \in [1/4, 1/2]$. Then with the above $k$, we know

$$k\mu\eta \geq \frac{2L}{\mu\alpha}\mu\eta = \frac{2}{\alpha}L\eta \geq \frac{1}{2\alpha}.$$

Hence

$$\frac{3k\mu\eta}{3k\mu\eta + 2} \geq \frac{3}{3 + 4\alpha}. \tag{A.35}$$

It then follows that

$$\sum_{t=0}^{T-1} \exp\left\{ -\alpha \frac{3k\mu\eta}{3k\mu\eta + 2} t \right\} \leq \frac{1}{1 - e^{-3\alpha/(3+4\alpha)}} \asymp 1.$$

Also, since $\mu\eta \leq \mu/2L \leq 1$, we can use $1 - e^{-x} \geq x/2$ for $x \in (0, 1]$ and get

$$\sum_{k'=0}^{k-1} e^{-\frac{3}{2}\mu\eta k'} = \frac{1 - e^{-\frac{3}{2}\mu\eta k}}{1 - e^{-\frac{3}{2}\mu\eta}} \leq \frac{1}{1 - e^{-\mu\eta}} \leq \frac{2}{\mu\eta}.$$

Plugging these into (A.34) yields the bound for the second term

$$\frac{L\alpha}{2} \sum_{t=0}^{T-1} \exp\left\{ -\alpha \frac{3k\mu\eta}{3k\mu\eta + 2} T \right\} \sum_{k'=0}^{k-1} \exp\{-\frac{3}{2}\mu\eta k'\} \frac{2\eta^2 L}{b} \mathbb{E}\left[F_S(\mathbf{w}_S)\right] \lesssim \frac{L\alpha}{2} \frac{2}{\mu\eta} \frac{2\eta^2 L}{b} \mathbb{E}\left[F_S(\mathbf{w}_S)\right]$$

$$\lesssim \frac{L^2\eta}{\mu b} \mathbb{E}\left[F_S(\mathbf{w}_S)\right].$$

Since $\eta = \frac{\mu b}{2L^2(b+1)}$, this simplifies to

$$\frac{L\alpha}{2} \sum_{t=0}^{T-1} \exp\left\{ -\alpha \frac{3k\mu\eta}{3k\mu\eta + 2} t \right\} \sum_{k'=0}^{k-1} \exp\{-\frac{3}{2}\mu\eta k'\} \frac{2\eta^2 L}{b} \mathbb{E}\left[F_S(\mathbf{w}_S)\right] \lesssim \frac{1}{2(b+1)} \mathbb{E}\left[F_S(\mathbf{w}_S)\right] \lesssim \mathbb{E}\left[F_S(\mathbf{w}_S)\right].$$

$$\tag{A.36}$$

For the first term, together with (A.35), our choice of $T$ ensures

$$\frac{L}{2} \exp\left\{ -\alpha \frac{3k\mu\eta}{3k\mu\eta + 2} T \right\} \mathbb{E}\left[\|\mathbf{w}_0 - \mathbf{w}_S\|^2\right] \lesssim \frac{L}{n} \mathbb{E}\left[\|\mathbf{w}_0 - \mathbf{w}_S\|^2\right]. \tag{A.37}$$

Combining (A.36) and (A.37) gives the final result. $\qquad\square$

We now state and prove the generalization bound.

**Lemma 12** (Generalization Gap of Lookahead: Strongly Convex Case). *Suppose the assumptions in Theorem 5 hold. Let $\mathbf{w}_T$ be the final output of Lookahead optimizer. By setting the learning rate $\eta = \frac{b\mu}{2L^2(b+1)}$, we have*

$$\mathbb{E}[F(\mathbf{w}_T) - F_S(\mathbf{w}_T)] \lesssim \frac{1}{n\mu} + \frac{1}{n^2}\mathbb{E}[\|\mathbf{w}_0 - \mathbf{w}_S\|^2] + \frac{1}{nL}\mathbb{E}[F_S(\mathbf{w}_S)].$$

*Proof of Lemma 12.* We now assume the constant step size $\eta_{\tau,t} = \eta$. Let $\mathbf{w}_S = \arg\min_{\mathbf{w}\in\mathcal{W}} F_S(\mathbf{w})$. We denote $B_{k,t} = \{z_{i_{k,t}^{(1)}}, \ldots, z_{i_{k,t}^{(b)}}\}$ and $f(\mathbf{v}; B_{k,t}) = \frac{1}{b}\sum_{j=1}^{b} f(\mathbf{v}; z_{i_{k,t}^{(j)}})$. We can hence reformulate the minibatch SGD update as

$$\mathbf{v}_{\tau+1,t} = \mathbf{v}_{\tau,t} - \eta\nabla f(\mathbf{v}_{\tau,t}; B_{\tau,t}). \tag{A.38}$$

By the strong convexity of $f$,

$$\mathbb{E}[\|\mathbf{v}_{\tau+1,t} - \mathbf{w}_S\|_2^2] = \mathbb{E}[\|\mathbf{v}_{\tau,t} - \eta\nabla f(\mathbf{v}_{\tau,t}; B_{\tau,t}) - \mathbf{w}_S\|_2^2]$$

$$= \mathbb{E}[\|\mathbf{v}_{\tau,t} - \mathbf{w}_S\|_2^2] - 2\eta\mathbb{E}[\langle\mathbf{v}_{\tau,t} - \mathbf{w}_S, \nabla F_S(\mathbf{v}_{\tau,t})\rangle] + \eta^2\mathbb{E}[\|\nabla f(\mathbf{v}_{\tau,t}; B_{\tau,t})\|_2^2]$$

$$\leq (1 - \mu\eta_{\tau,t})\mathbb{E}[\|\mathbf{v}_{\tau,t} - \mathbf{w}_S\|_2^2] - 2\eta\mathbb{E}[F_S(\mathbf{v}_{\tau,t}) - F_S(\mathbf{w}_S)] + \eta^2\mathbb{E}[\|\nabla f(\mathbf{v}_{\tau,t}; B_{\tau,t})\|_2^2]. \tag{A.39}$$

For the last term, we bound it using (A.16) and get

$$\mathbb{E}[\|\mathbf{v}_{\tau+1,t} - \mathbf{w}_S\|_2^2] \leq (1 - \mu\eta)\mathbb{E}[\|\mathbf{v}_{\tau,t} - \mathbf{w}_S\|_2^2] - \left(2\eta - \frac{2L\eta^2(b+1)}{b}\right)\mathbb{E}\left[F_S(\mathbf{v}_{\tau,t}) - F_S(\mathbf{w}_S)\right] + \frac{2L\eta^2\mathbb{E}\left[F_S(\mathbf{w}_S)\right]}{b}.$$

For $\eta = \frac{b\mu}{2L^2(b+1)} \leq \frac{b}{2L(b+1)}$, we have

$$\mathbb{E}[\|\mathbf{v}_{\tau+1,t} - \mathbf{w}_S\|_2^2] \leq (1 - \mu\eta)\mathbb{E}[\|\mathbf{v}_{\tau,t} - \mathbf{w}_S\|_2^2] - \eta\mathbb{E}[F_S(\mathbf{v}_{\tau,t}) - F_S(\mathbf{w}_S)] + \frac{2L\eta^2\mathbb{E}\left[F_S(\mathbf{w}_S)\right]}{b}.$$

We multiply both sides by $\left(1 - \frac{\alpha}{2}\right)^{T-t}(1-\mu\eta/2)^{k-\tau}$ and get

$$\left(1 - \frac{\alpha}{2}\right)^{T-t}(1-\mu\eta/2)^{k-\tau}\mathbb{E}[\|\mathbf{v}_{\tau+1,t} - \mathbf{w}_S\|_2^2] \le \left(1 - \frac{\alpha}{2}\right)^{T-t}(1-\mu\eta/2)^{k-\tau+1}\mathbb{E}[\|\mathbf{v}_{\tau,t} - \mathbf{w}_S\|_2^2] -$$

$$\left(1 - \frac{\alpha}{2}\right)^{T-t}\eta(1-\mu\eta/2)^{k-\tau}\mathbb{E}[F_S(\mathbf{v}_{\tau,t}) - F_S(\mathbf{w}_S)] + \frac{2L(1 - \frac{\alpha}{2})^{T-t}(1-\mu\eta/2)^{k-\tau}\eta^2\mathbb{E}[F_S(\mathbf{w}_S)]}{b}.$$

By taking a summation of the above inequality, we have

$$\sum_{t=1}^{T}\left(1 - \frac{\alpha}{2}\right)^{T-t}\sum_{\tau=0}^{k-1}\eta_{\tau,t}(1-\mu\eta/2)^{k-\tau}\mathbb{E}[F_S(\mathbf{v}_{\tau,t}) - F_S(\mathbf{w}_S)]$$

$$\le \sum_{t=1}^{T}\left(1 - \frac{\alpha}{2}\right)^{T-t}(1-\mu\eta/2)^{k+1}\mathbb{E}[\|\mathbf{w}_{t-1} - \mathbf{w}_S\|_2^2] + 2L\sum_{t=1}^{T}\left(1 - \frac{\alpha}{2}\right)^{T-t}\sum_{\tau=0}^{k}\frac{(1-\mu\eta/2)^{k-\tau}\eta^2\mathbb{E}[F_S(\mathbf{w}_S)]}{b}$$

$$\le \frac{1}{2}\sum_{t=1}^{T}\left(1 - \frac{\alpha}{2}\right)^{T-t}\mathbb{E}[\|\mathbf{w}_{t-1} - \mathbf{w}_S\|_2^2] + 2L\sum_{t=1}^{T}\left(1 - \frac{\alpha}{2}\right)^{T-t}\sum_{\tau=0}^{k}\frac{(1-\mu\eta/2)^{k-\tau}\eta^2\mathbb{E}[F_S(\mathbf{w}_S)]}{b},$$

$$\tag{A.40}$$

where we have used Eq. (A.24). We first look at the first term of Eq. (A.40). By (A.32), we have

$$\mathbb{E}[\|\mathbf{w}_{t-1} - \mathbf{w}_S\|_2^2] \le \rho^t\mathbb{E}[\|\mathbf{w}_0 - \mathbf{w}_S\|^2] + \alpha\sum_{t'=0}^{t-2}\rho^{t'}\sum_{k'=0}^{k-1}\left(1 - \frac{3}{2}\mu\eta\right)^{k'}\frac{2L\eta^2\mathbb{E}[F_S(\mathbf{w}_S)]}{b}$$

$$\lesssim \frac{1}{n}\mathbb{E}[\|\mathbf{w}_0 - \mathbf{w}_S\|^2] + \frac{1}{L}\mathbb{E}[F_S(\mathbf{w}_S)].$$

where the last inequlity follows from the result of (A.30) and the fact that $\mathbb{E}[F_S(\mathbf{w}_t) - F_S(\mathbf{w}_S)] \le \frac{L}{2}\mathbb{E}[\|\mathbf{w}_t - \mathbf{w}_S\|^2] \lesssim \frac{L}{n}\mathbb{E}[\|\mathbf{w}_0 - \mathbf{w}_S\|^2] + \mathbb{E}[F_S(\mathbf{w}_S)]$. Together with the summation, we have

$$\frac{1}{2}\sum_{t=1}^{T}\left(1 - \frac{\alpha}{2}\right)^{T-t}\mathbb{E}[\|\mathbf{w}_{t-1} - \mathbf{w}\|_2^2] \lesssim \frac{1}{2}\sum_{t=1}^{T}\left(1 - \frac{\alpha}{2}\right)^{T-t}\left(\frac{1}{n}\mathbb{E}[\|\mathbf{w}_0 - \mathbf{w}_S\|^2] + \frac{1}{L}\mathbb{E}[F_S(\mathbf{w}_S)]\right)$$

$$\le \frac{1}{2}\frac{1}{1-(1-\frac{\alpha}{2})}\left(\frac{1}{n}\mathbb{E}[\|\mathbf{w}_0 - \mathbf{w}_S\|^2] + \frac{1}{L}\mathbb{E}[F_S(\mathbf{w}_S)]\right)$$

$$\lesssim \frac{1}{n}\mathbb{E}[\|\mathbf{w}_0 - \mathbf{w}_S\|^2] + \frac{1}{L}\mathbb{E}[F_S(\mathbf{w}_S)]. \tag{A.41}$$

For the second term of (A.40), by Eq. (A.28) and $\eta \le \frac{\mu}{2L^2}$,

$$2L\sum_{t=1}^{T}\left(1 - \frac{\alpha}{2}\right)^{T-t}\sum_{\tau=0}^{k}\frac{(1-\mu\eta/2)^{k-\tau}\eta^2\mathbb{E}[F_S(\mathbf{w}_S)]}{b} \le \frac{\mu}{\alpha L}\sum_{\tau=0}^{k}\frac{(1-\mu\eta/2)^{k-\tau}\eta\mathbb{E}[F_S(\mathbf{w}_S)]}{b}$$

$$\lesssim \frac{\mathbb{E}[F_S(\mathbf{w}_S)]}{\alpha L}. \tag{A.42}$$

We fix the outer-loop learning rate $\alpha$ and combine Eq. (A.41) and Eq. (A.42) to obtain

$$\sum_{t=1}^{T}\left(1 - \frac{\alpha}{2}\right)^{T-t}\sum_{\tau=0}^{k-1}\eta(1-\mu n/2)^{k-(\tau+1)}\mathbb{E}[F_S(\mathbf{v}_{\tau,t}) - F_S(\mathbf{w}_S)] \lesssim \frac{1}{n}\mathbb{E}[\|\mathbf{w}_0 - \mathbf{w}_S\|^2] + \frac{1}{L}\mathbb{E}[F_S(\mathbf{w}_S)].$$

$$\tag{A.43}$$

Recall from Eq. (5.4), we denote $S_T$:

$$S_T = \sum_{t'=1}^{T}\left(1 - \frac{\alpha}{2}\right)^{T-t}\sum_{j=0}^{k-1}\eta_{j,t'}\sqrt{\mathbb{E}[F_S(\mathbf{v}_{j,t'})]}(1-\mu n/2)^{k-(\tau+1)}.$$

We use the inequality $\sqrt{x} \le (1+x)/2$ for non-negative $x$. This gives:

$$S_T \le \frac{1}{2}\sum_{t'=1}^{T}\left(1 - \frac{\alpha}{2}\right)^{T-t}\sum_{j=0}^{k-1}\eta_{j,t'}\left(1 + \mathbb{E}[F_S(\mathbf{v}_{j,t'})]\right)(1-\mu n/2)^{k-(\tau+1)}.$$

We split this into two parts,

$$S_T \le \frac{1}{2}\underbrace{\left[\sum_{t'=1}^{T}\left(1 - \frac{\alpha}{2}\right)^{T-t}\sum_{j=0}^{k-1}\eta_{j,t'}(1-\mu n/2)^{k-(\tau+1)}\right]}_{\text{Part A}} + \frac{1}{2}\underbrace{\left[\sum_{t'=1}^{T}\left(1 - \frac{\alpha}{2}\right)^{T-t}\sum_{j=0}^{k-1}\eta_{j,t'}\mathbb{E}[F_S(\mathbf{v}_{j,t'})](1-\mu n/2)^{k-(\tau+1)}\right]}_{\text{Part B}}.$$

We bound each part:

Part A: This part is bounded using the result from Eq. (A.28). The identity shows that for each outer step $t'$, the inner sum over $j$ is bounded by $2/\mu$. Summing over $T$ outer steps yields:

$$\frac{1}{2}\sum_{t'=1}^{T}\left(1-\frac{\alpha}{2}\right)^{T-t}\sum_{j=0}^{k-1}\eta_{j,t'}(1-\mu n/2)^{k-(\tau+1)} \lesssim \frac{1}{\mu}. \tag{A.44}$$

Part B: Notice that

$$\sum_{t=1}^{T}\left(1-\frac{\alpha}{2}\right)^{T-t}\sum_{\tau=0}^{k-1}\eta(1-\mu n/2)^{k-(\tau+1)}\mathbb{E}[F_S(\mathbf{v}_{\tau,t})]$$

$$=\sum_{t=1}^{T}\left(1-\frac{\alpha}{2}\right)^{T-t}\sum_{\tau=0}^{k-1}\eta(1-\mu n/2)^{k-(\tau+1)}\mathbb{E}[F_S(\mathbf{v}_{\tau,t})-F_S(\mathbf{w}_S)]+\sum_{t=1}^{T}\left(1-\frac{\alpha}{2}\right)^{T-t}\sum_{\tau=0}^{k-1}\eta(1-\mu n/2)^{k-(\tau+1)}\mathbb{E}[F_S(\mathbf{w}_S)]$$

$$\lesssim \frac{1}{n}\mathbb{E}[\|\mathbf{w}_0-\mathbf{w}_S\|^2]+\frac{1}{L}\mathbb{E}[F_S(\mathbf{w}_S)]. \tag{A.45}$$

Combining (A.44) and (A.45) we have:

$$\frac{1}{n}\sum_{i=1}^{n}\mathbb{E}\left[\|\mathbf{w}_T-\mathbf{w}_T^{(i)}\|_2\right] \lesssim \frac{1}{n\mu}+\frac{1}{n^2}\mathbb{E}[\|\mathbf{w}_0-\mathbf{w}_S\|^2]+\frac{1}{nL}\mathbb{E}[F_S(\mathbf{w}_S)]. \tag{A.46}$$

By Lemma 1 (a), (A.46) implies

$$\mathbb{E}[F(\mathbf{w}_T)-F_S(\mathbf{w}_T)] \lesssim \frac{1}{n\mu}+\frac{1}{n^2}\mathbb{E}[\|\mathbf{w}_0-\mathbf{w}_S\|^2]+\frac{1}{nL}\mathbb{E}[F_S(\mathbf{w}_S)]. \tag{A.47}$$

The proof is completed. $\square$

*Proof of Theorem 6.* Note that for $\alpha \leq \frac{b\mu}{2\ln 2(b+1)L}$, we have

$$\eta=\frac{b\mu}{2L^2(b+1)}\geq\frac{\ln 2}{L}\alpha=\frac{2\ln 2}{\mu}\frac{\alpha\mu}{2L}=\frac{2\ln 2}{\mu k}$$

Which satisfy the required condition in theorem 5. We now combine the results of lemma 12 and lemma 11 together and get

$$\mathbb{E}[F(\mathbf{w}_T)-F(\mathbf{w}^*)] \lesssim \frac{1}{n\mu}+\left(\frac{1}{nL}+1\right)\mathbb{E}[F_S(\mathbf{w}_S)]+\left(\frac{1}{n^2}+\frac{L}{n}\right)\mathbb{E}[\|\mathbf{w}_0-\mathbf{w}_S\|^2]. \tag{A.48}$$

for $k=\frac{2L}{\alpha\mu}$, and $T \asymp \log(\mu n)$. This completes the proof. $\square$

