# OpenReview forum: "Generalization and Optimization of SGD with Lookahead"
_ICLR.cc/2026/Conference — ICLR 2026 Conference Withdrawn Submission_

### Official Review · Reviewer_RhKm · 2025-10-20

**Soundness:** 2
**Presentation:** 3
**Contribution:** 2
**Rating:** 4
**Confidence:** 4

**Summary:**

This paper conducts a comprehensive study of the generalization analysis of Lookahead with SGD using tools from algorithmic stability. Compared to previous results (Zhou et al., 2021), this paper made several significant improvements for the (strongly-) convex problems. However, this paper does not study the non-convex setting.

**Strengths:**

Lookhead is an important optimizer that turns out to be very effective in training large-scale neural networks. This work attempts to connect its generalization performance with the optimization process. Using standard tools from algorithmic stability, on one hand, this paper removes the global Lipshitz assumption. On other hand, the authors also demonstrate a linear speedup w.r.t. the batch size, which is lacking in Zhou et al., 2021.

**Weaknesses:**

This paper is well-organized and easy to follow. However, I still have a few comments as follows:

 - In line 264, the authors state that **our stability bounds become progressively tighter throughout the optimization process**. Recall that RHS of Equation (5.1), this term is the sum of $\sqrt{E[F_S(v_{j, h})]}$ over $h, j$. As the optimization continues, more terms are added to this term. So, I do not understand how the bound becomes tighter with training process. Please clarify.
 - In line 268, the claim that **The term $1/nb$ shows that increasing the minibatch size improves stability** is questionable. According to Equation (5.2), when increasing the batch size, we do not know how the term $E[F_S(v_{j, h})]$ changes. Please clarify.
 - In Corollary 4, I am curious that why $\alpha$ is missing, it's an important hyper-parameter of Lookahead as well. Moreover, I am wondering $\eta=\frac{b}{\sqrt{nF(w^\ast)}}$ in line 308 is a reasonable choice. Generally, $F(w^\ast)$ is a very small value.
 - At last, I would like to see a toy example that could verify the theoretical implications.

**Questions:**

see Weaknesses.

---

### Official Review · Reviewer_UcaQ · 2025-10-27

**Soundness:** 3
**Presentation:** 2
**Contribution:** 2
**Rating:** 2
**Confidence:** 4

**Summary:**

This paper analyzes the generalization and stochastic optimization performances of the Lookahead algorithm, a double-loop optimizer effectively used in deep learning, with minibatch SGD used for inner loop optimization. Different from the exisitng analyses that typically rely on the global Lipschitz continuity of loss functions, the current analysis uses on-average model stability to derive generalization bounds for smooth and (strongly) convex problems without restrictive Lipschitz assumptions. The main results include a set of optimistic generalization and excess risk bounds tied to empirical risk, and a demonstration of linear speedup with respect to batch size in the smooth and convex setting.

**Strengths:**

1. The most notable advantage of the proposed analysis lies in that it relaxes the restrictive global Lipschitzness assumption commonly used in the existing generalization analyses of the Lookahead optimizer. This enables valid generalization bounds even for high-dimensional problems with unbounded gradients, expanding the range of theoretical applicability of the optimizer.

2. Under the smoothness assumption, the paper establishes some novel optimistic stability bounds that depend on the algorithm’s empirical risk (which decreases during training) rather than worst-case global constants. This makes the bounds progressively tighter as optimization proceeds, offering a more refined and practical characterization of stability compared to prior work。

3. By combining the derived optimistic generalization bounds with some prioer optimization bounds, the paper respectively proves $O(1/n + F^{\star})$ or $O(\sqrt{F^{\star}/n})$ excess risk rates for convex problems, and $O(1/(nμ))$ for $μ$-strongly convex problems. Also, it demonstrates a linear speedup with batch size in convex settings—larger batches reduce required iterations proportionally. These findings outperform prior bounds (e.g., Zhou et al., 2021) by capturing low-noise acceleration and batch size advantages.

**Weaknesses:**

My main concern is about the novely of analysis, significance of results, and quality of presentation/organization, as specified below.

1. The stability and generalization analysis is incrementally novel, mostly extending existing proof techniques for the algorithmic stability of SGD (e.g., Lei & Ying, 2020)  without any particularly new insights or tools alongside developed for Lookahead. Plus, while the core results are intuitive and interesting, they do not explicitly reveal any theoretical advantage of Lookahead over SGD. It is thus evaluated that the current analyses and results are unlikely to generate substantial impact on the theoretical understanding of Lookahead, neither in generalization nor in optimization.

2. Concerning the benefit of minibatching for performance speedup, it is well-known that in the considered smooth optimization setting, the minibatch size $b$ indeed works for linearly reducing the iteration complexity (via naive variance reduction). Since the generalization rates of SGD-type algorithms usually scale proportionally with respect to the rounds of iteraiton (see, e.g., Hardt et al. 2016, or Theorem 2 in this paper) , the reduced iteration complexity naturally leads to faster generalization. It is thus a bit misleading to claim (say, in Line 268) solely based on the stability bounds in Theorem 2 that increasing the minibatch size improves stability. As a matter of fact, it can be clearly seen in the bounds of Theorem 2 that the minibatch size $b$ does not appear in the dominant terms.

3. Continuing that above comment, while the paper demonstrates a linear speedup with respect to batch size in the convex setting, it explicitly acknowledges that this key advantage does not extend to strongly convex problems. This gap means the algorithm’s scalability benefits are incomplete, as strongly convex tasks cannot leverage batch size increases to accelerate training.

4. The paper organization has room for improvement, in the sense that it suffers from an unbalanced section distribution: excessive sections (~ 4 out of 7.5 pages) have been dedicated to background, preliminaries, and related work, while only Section 5 (~ 3 pages) focuses on the core results. This imbalance scatters supporting context across too many sections, forcing readers to navigate through non-essential content to access main points. In the meanwhile, some key insights like impacts of minibatch size on convex/strongly convex stability are buried in scattered remarks, lacking a unified summary.

5. It is claimed in the conclusion that the optimistic rates of Lookahead in the convex case can be achieved without prior knowledge of the optimal risk. This is inaccurate IMO as Corollary 4 obviously needs the knowledge of $F^*$ in the choices of hyperparameters like $\eta$, $\gamma$, $b$.

**Questions:**

In view of the``train faster, generalize better'' phnomenon of convex SGD (Hardt et al. 2016), and since the linear speedup of minibatch SGD is achievable for smooth and strongly convex optimization, I am wondering is it possible to show linear speedup of Lookahead in excess risk rates for strongly convex problems?

---

### Official Review · Reviewer_6Kcb · 2025-10-28

**Soundness:** 3
**Presentation:** 3
**Contribution:** 2
**Rating:** 4
**Confidence:** 4

**Summary:**

This paper focuses on the generalization of Lookahead optimizers.
Motivated by the existing analysis's Lipschitz assumption and looseness, the paper applies on-average stability to prove stability-based generalization bounds in convex and strongly convex problems that features no Lipschitzness dependence and dependence on empirical risks along the optimization trajectories.
The bounds also reveal that compared to standard SGD, Lookahead optimizers can make the trajectory more stable by introducing an $\alpha$ factor in the bounds, can use larger batch size to linearly accelerate optimization while achieving optimal rates and can achieve optimal rates without prior knowledge of $F(w^*)$.

**Strengths:**

- The new bounds remove the requirement on Lipschitzness of existing results, which is not satisfied in many (deep) machine learning problems. The bounds are also more adaptive and data-dependent.
- The new bounds reveal benefits for generalization of Lookahead optimizers compared to standard SGD.
- In the technique perspective, the on-average stability framework from Lei & Ying (2020) is adapted at least on the following aspects: Outer loop (Eq(4.1)) that is specific to Lookahead optimizers, and $>1$ batch size to reveal the role of batch size in Lookahead optimizers.

**Weaknesses:**

- Some claims are not fully supported: In the Conclusion, the authors claim that in the convex case, Lookahead optimizers can achieve optimal rates without access to $F(w^\*)$. However, in Corollary 4, for  $F(w^\*) \ge 1/n$ case, $\eta$ and the upperbound of batch size $b$ depends on $F(w^\*)$. Since $1/n \to 0 \le F(w^*)$ as $n \to \infty$, this case cannot be ignored for the asymptotic rate.
- In Sec 5.1, the roles of Lookahead hyperparameters explicitly revealed by the bounds are discussed. Nevertheless, aside from explicit roles, they seem to have a lot of implicit impacts. For example, when $\alpha$ is decreased, the outer-loop makes slower updates and slows the decrease of empirical risks. Such implicit impacts seem to contradict with the explicit roles, which is not discussed. What is the net effect of these impacts? Do the other hyperparameters also have some implicit impacts?
- In Corollary 4, the hyperparameters $\alpha$ that is critical to Lookahead optimizers are hidden in $\lesssim$ and not optimized. Will they reveal more advantages of Lookahead optimizers over standard SGD?
- Techniques are rather standard with specific adaptation. Technical contribution is limited.

**Questions:**

- In Theorem 2, the upperbounds for the two stabilities has different dependence on hyperparameters. For example, the $\ell_1$ stability bound does not depends on the batch size $b$. the $\ell_2$ depends on the latter without dependence on the former. What is the (intuitive) reason behind such difference in dependence?
- Can one take $\alpha=1$? If so, the Lookahead optimizer with $\alpha=1$ essentially fall back to SGD. If the results (eg, Theorem/Corollary 2-4) still hold under $\alpha=1$, then the claimed Lookahead-specific properties (eg, linear acceleration by increasing batch size, no need for knowledge on $F(w^*)$) can already be observed by SGD. Is this extension valid? If so, what is the advantage of Lookahead?  This question is connected to my Weaknesses 3, where $\alpha$ is not optimized, ie, Lookahead is given up to some extent and the Corollary cannot differentiate Lookahead and SGD.

---

### Official Review · Reviewer_BdKj · 2025-11-01

**Soundness:** 3
**Presentation:** 3
**Contribution:** 2
**Rating:** 4
**Confidence:** 3

**Summary:**

This paper provides a theoretical analysis of the Lookahead optimizer from the perspective of algorithmic stability and generalization. The authors remove the restrictive Lipschitz continuity assumption on the loss function and establish new on-average model stability bounds for both convex and strongly convex settings. The paper derives optimistic generalization bounds depending on the empirical risk, and shows that Lookahead can achieve optimal excess risk rates of  𝑂(1/𝑛) for convex problems and 𝑂(1/(𝑛𝜇)) for strongly convex problems, with linear speedup in batch size.

**Strengths:**

1. The paper is well organized and clearly written, with precise mathematical definitions and complete proofs.
2. The removal of the Lipschitz assumption broadens the theoretical scope of existing analyses.
3.  The use of on-average model stability leads to optimistic, data-dependent generalization bounds.

**Weaknesses:**

1. The analysis mainly extends existing results [1] to the Lookahead setting.
The proofs closely follow the standard framework used for SGD and Local SGD, with a few new theoretical ideas.
2. While the paper quantifies stability improvements, it does not provide a deeper understanding of why the Lookahead interpolation enhances generalization. The mechanism-level interpretation (e.g., implicit regularization, variance reduction) remains unexplored.
3. The work is purely theoretical; even simple simulations demonstrating the predicted linear speedup or empirical stability would improve the impact.
4. The analysis is restricted to convex and strongly convex cases. Given that Lookahead is primarily used in non-convex deep learning, the theoretical relevance is limited for modern practice.

[1] Yunwen Lei and Yiming Ying. Fine-grained analysis of stability and generalization for stochastic gradient descent. In International Conference on Machine Learning, pp. 5809–5819, 2020.

**Questions:**

1. Can the proposed stability analysis be extended to non-convex setting?
2. How does the α interpolation parameter affect implicit bias or sharpness in deep networks?
3. It should be better to incorporate empirical results to strengthen the claims.

---

### Note · Authors · 2025-11-19

**Comment:**

Dear Reviewers,

We sincerely thank you for taking the time to provide us with so many valuable comments. We are delighted that the reviewers recognized the fundamental motivation and value of this paper. On the other hand, the fact that we did not receive recommendations for acceptance from any reviewers made us realize that this paper has room for improvement. Therefore, we have decided to withdraw this paper and strengthen it based on your comments before resubmitting it to another conference. Finally, we would like to express our sincere gratitude once again for your detailed and insightful comments.

Best regards,
Authors

**Withdrawal Confirmation:**

I have read and agree with the venue's withdrawal policy on behalf of myself and my co-authors.